# Pitx2c orchestrates embryonic axis extension via mesendodermal cell migration

Michelle M Collins[1]*, Hans-Martin Maischein[1], Pascale Dufourcq[2], Marine Charpentier[3], Patrick Blader[2], Didier YR Stainier[1]*

[1]Department of Developmental Genetics, Max Planck Institute for Heart and Lung Research, Bad Nauheim, Germany; [2]Centre de Biologie du Développement, Centre de Biologie Intégrative, Université Toulouse III - Paul Sabatier, CNRS, Toulouse, France; [3]Muséum National d'Histoire Naturelle, Inserm U1154, CNRS, Paris, France

**Abstract** Pitx2c, a homeodomain transcription factor, is classically known for its left-right patterning role. However, an early wave of *pitx2* expression occurs at the onset of gastrulation in several species, indicating a possible earlier role that remains relatively unexplored. Here we show that in zebrafish, maternal-zygotic (MZ) *pitx2c* mutants exhibit a shortened body axis indicative of convergence and extension (CE) defects. Live imaging reveals that MZ*pitx2c* mutants display less persistent mesendodermal migration during late stages of gastrulation. Transplant data indicate that Pitx2c functions cell non-autonomously to regulate this cell behavior by modulating cell shape and protrusive activity. Using transcriptomic analyses and candidate gene approaches, we identify transcriptional changes in components of the chemokine-ECM-integrin dependent mesendodermal migration network. Together, our results define pathways downstream of Pitx2c that are required during early embryogenesis and reveal novel functions for Pitx2c as a regulator of morphogenesis.
DOI: https://doi.org/10.7554/eLife.34880.001

*For correspondence:
michelle.collins@mpi-bn.mpg.de (MMC);
Didier.Stainier@mpi-bn.mpg.de (DYRS)

## Introduction

The process of gastrulation reorganizes cells of the embryonic blastula into the three primary germ layers through a series of complex cell movements. Gastrulation movements in zebrafish, including epiboly, ingression, convergence, and extension, are driven by many cell behaviors such as directed cell migration, mediolateral cell intercalation, and cell shape changes (reviewed by *Solnica-Krezel and Sepich, 2012*; *Tada and Heisenberg, 2012*; *Williams and Solnica-Krezel, 2017*). Morphogenetic movements not only shape the embryo during gastrulation, but are also crucial for its patterning by causing the cells to be exposed to particular signaling cues leading them to acquire specific cell fates.

One cue that is critical during gastrulation is Nodal signaling, which drives cells towards mesendodermal fates. Nodal signaling activates many targets including itself and its own inhibitors, such as Lefty, to ensure tight spatial and temporal control (reviewed by *Shen, 2007*; *Schier, 2009*). In the context of left-right patterning, Nodal activates the expression of the transcription factor gene *pitx2c* (*Bisgrove et al., 1999*; *Essner et al., 2000*). The Nodal-Lefty-Pitx2 cassette is highly conserved, as Pitx2 is a key player during asymmetric morphogenesis from echinoderms to chordates (*Levin et al., 1995*; *Piedra et al., 1998*; *Ryan et al., 1998*; *Yoshioka et al., 1998*; *Lu et al., 1999*; *Boorman and Shimeld, 2002*; *Duboc et al., 2005*). Animal models have also revealed important functions for Pitx2 during craniofacial, cardiac, and pituitary development. Three major *Pitx2* isoforms are produced in mouse, chick, and frog; *Pitx2a* and *Pitx2b* are generated by alternate splicing whereas *Pitx2c* uses a different promoter (*Schweickert et al., 2000*; *Cox et al., 2002*). In contrast,

only two isoforms have been identified in zebrafish, *pitx2a* and *pitx2c* (*Essner et al., 2000*). Antisense morpholinos designed to target both isoforms have been reported to affect embryonic development (*Bohnsack et al., 2012*; *Liu and Semina, 2012*) resulting in craniofacial and ocular defects reminiscent of the Axenfeld-Rieger syndrome phenotypes caused by mutations in human *PITX2* (*Semina et al., 1996*; *Priston et al., 2001*; *Lines et al., 2002*). Specific knockdown of *pitx2c* in zebrafish affects habenular nuclei asymmetry by modulating parapineal cell number (*Garric et al., 2014*). More recently, zebrafish *pitx2* mutants have been generated. Mutations that lead to a truncation of the homeodomain and affect both *pitx2a* and *pitx2c* cause eye, craniofacial, and tooth defects (*Ji et al., 2016*; *Hendee et al., 2018*). These defects were not observed in *pitx2c*-specific mutants (*Ji et al., 2016*); however only male homozygous mutants were recovered at the adult stage, precluding simple analysis of maternal zygotic phenotypes.

Several lines of evidence have revealed roles for Pitx2 during gastrulation. In zebrafish, cells in both the blastoderm margin and the prechordal plate are exposed to high levels of Nodal signaling. Accordingly, *pitx2* expression is observed in the blastoderm margin at the onset of gastrulation (*Faucourt et al., 2001*) and becomes highly enriched in the anterior mesendoderm which subsequently forms the prechordal plate. Intriguingly, this early wave of expression that is coincident with the onset of gastrulation is observed in multiple species. Mouse transcriptomic analyses detect *Pitx2* expression at E6.25 (*Mitiku and Baker, 2007*), around the time that the primitive streak forms. Similarly, *pitx2* expression is detected in the early gastrula of both *Xenopus laevis* (*Ding et al., 2017*) and *X. tropicalis* (*Blitz et al., 2017*). Previous studies in *X. laevis* have reported that *pitx2* expression can be induced by overexpression of the Nodal orthologue Xnr-1, and that *pitx2* overexpression partially phenocopies Nodal overexpression (*Campione et al., 1999*; *Faucourt et al., 2001*). Pitx2 was also identified in a screen for upstream regulators of *cVg1* expression in the posterior marginal zone of avian embryos (*Torlopp et al., 2014*); more recently, the observation that Pitx2 is a conserved determinant of axis formation has been described in rabbit (*Plöger and Viebahn, 2018*). Together, these data suggest that Pitx2 has conserved patterning and morphogenetic roles in the early embryo.

Here, we describe a role for Pitx2c during gastrulation movements in zebrafish. We show that in the absence of maternal and zygotic Pitx2c function, embryos display convergence and extension defects that are caused by aberrant cell behaviors. Through live imaging, we observed that MZ*pitx2c* mutants exhibit less persistent mesendodermal migration as well as fewer oriented cell divisions of deep cells during gastrulation. Transplantation data indicate that these phenotypes arise from cell non-autonomous functions of Pitx2c. Accordingly, using *pitx2c* gain- and loss-of-function experiments, we identify transcriptional changes in the chemokine ligand gene *cxcl12b,* an important driver of endodermal migration, and in the fibronectin receptor subunit gene *itgb1b,* providing mechanistic insight into the role of Pitx2c during gastrulation.

## Results

### Maternal and zygotic Pitx2c functions are required for embryonic development

The *pitx2* locus in zebrafish produces two transcripts, *pitx2a* and *pitx2c,* generated from distinct promoters (*Figure 1a*), and which encode identical homeodomains and C-termini but unique N-terminal activation domains (*Figure 1b*). It has been previously reported that no maternally contributed *pitx2* transcripts could be detected (*Faucourt et al., 2001*; *Ji et al., 2016*). However, using several different intron-spanning primers (*Figure 1a*) to detect the *pitx2c* isoform specifically, or primers flanking the homeodomain coding region common to both isoforms (exons 4 and 5), we were able to detect maternally-deposited *pitx2c* expression at very low levels by qPCR and sequencing (*Figure 1—figure supplement 1*). Following the onset of zygotic transcription, *pitx2c* expression becomes upregulated in a ring at the blastoderm margin (*Figure 1—figure supplement 1a*) in agreement with a previous report (*Faucourt et al., 2001*). At shield stage, *pitx2c* expression can be detected in the shield and margin (*Figure 1—figure supplement 1b*). By late gastrulation, expression becomes restricted to the prechordal plate mesoderm (*Figure 1—figure supplement 1c–e*). We also analyzed *pitx2c* expression at 80% epiboly in embryos that lack endoderm (*sox32* morphants), and found that

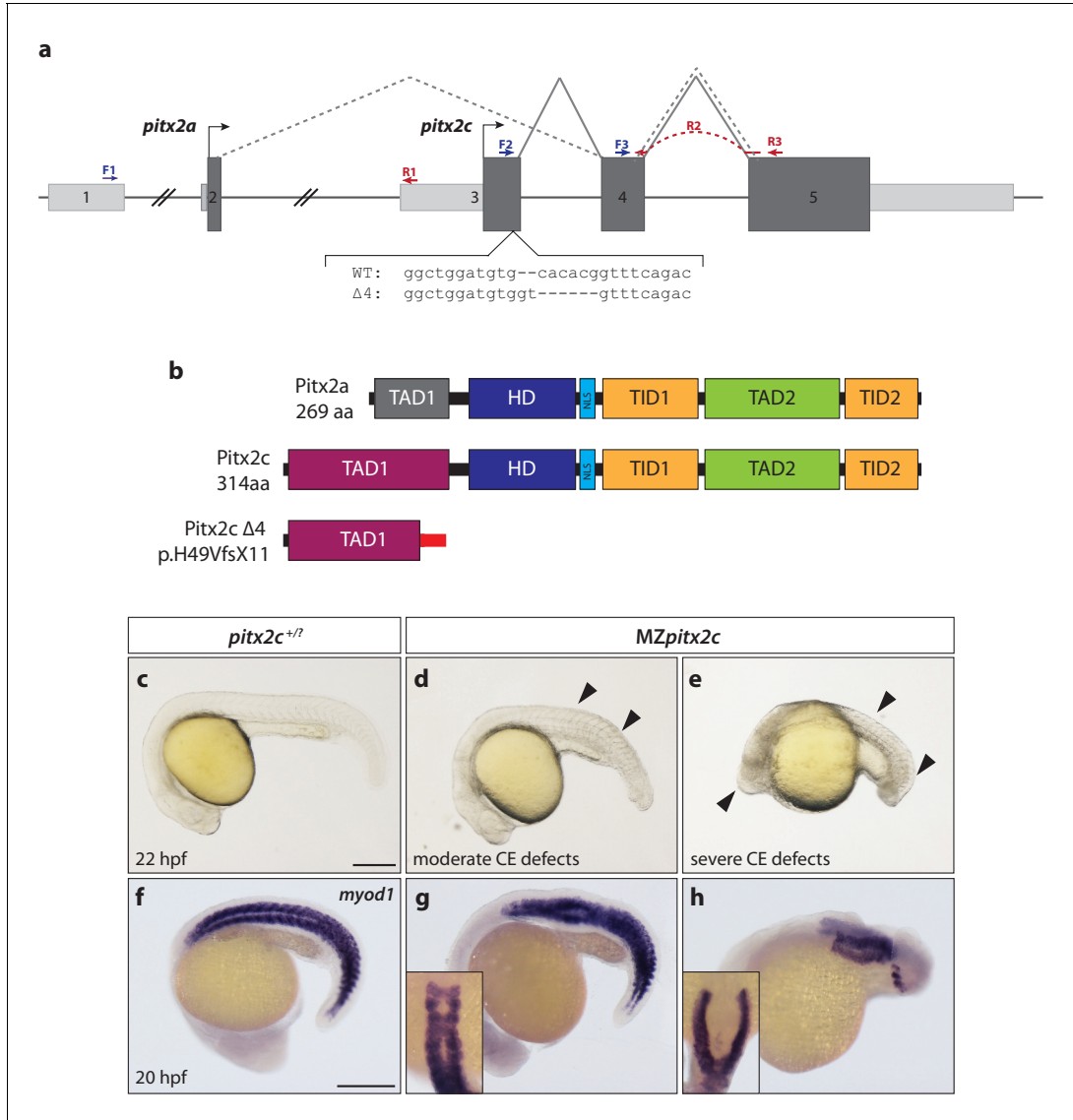

**Figure 1.** Maternal and zygotic Pitx2c function are required for embryonic development. (**a**) Schematic of the zebrafish *pitx2* locus including the sequence of the Δ4 allele. The location of forward (F) and reverse (R) primers used for qPCR are indicated in blue and red, respectively. The red dashed line for R2 indicates primer binding on the 3' end of exon 4 and the 5' end of exon 5. The gray dashed line indicates splicing of the *pitx2a* isoform, while the solid line indicates splicing of the *pitx2c* isoform. (**b**) Schematic of the domains of Pitx2a, Pitx2c, and the predicted truncated protein generated from the Δ4 transcript. (**c–h**) Phenotypes observed in MZ*pitx2c* mutant embryos obtained from homozygous mutant incrosses compared to wild-type embryos. MZ*pitx2c* mutant phenotypes are variable and can be classified into three groups: Class I embryos exhibit somite defects (**d, g**); Class II embryos exhibit severe axis shortening and other malformations suggestive of convergence and extension (CE) defects (**e, h**); Class III embryos exhibit no obvious morphological defects (not shown). HD: homeodomain; NLS, nuclear localization signal; TAD: transactivation domain; TID, transactivation inhibitory domain. Scale bars, 250 μm.

DOI: https://doi.org/10.7554/eLife.34880.002

The following figure supplements are available for figure 1:

**Figure supplement 1.** Maternal and zygotic expression of *pitx2c*.

DOI: https://doi.org/10.7554/eLife.34880.003

**Figure supplement 2.** Analysis of mRNA levels for *pitx2c* and related genes.

DOI: https://doi.org/10.7554/eLife.34880.004

**Figure supplement 3.** The *pitx2c* Δ4 allele encodes a non-functional protein.

DOI: https://doi.org/10.7554/eLife.34880.005

it appeared unchanged, suggesting that at this stage the majority of *pitx2c* expression is within the mesodermal lineage (*Figure 1—figure supplement 1f*).

To examine the function of Pitx2c, we generated a mutant allele of *pitx2c* using site-specific TALE nucleases. We recovered an allele containing a 4 bp deletion in the first exon of *pitx2c*, resulting in a premature stop codon and a predicted truncated protein lacking the homeodomain and downstream transcriptional inhibitory and activation domains (Δ4 allele; *pitx2c^ups6*; p.H49VfsX11) (*Figure 1a,b*; *Figure 1—figure supplement 2*). Non-Mendelian numbers of adult homozygous mutant animals were recovered and found to be viable and fertile. Homozygous mutant animals were incrossed to generate maternal zygotic *pitx2c* (MZ*pitx2c*) mutants. From these crosses, we were able to classify embryos at 24 hpf into three groups based on their phenotypes: Class I embryos (~30%) exhibited somites that were wider mediolaterally and narrower along the anterior-posterior axis, resulting in a shortened body axis (*Figure 1d,g*), Class II embryos (~15%) had severe axis defects and developmental abnormalities (*Figure 1e,h*), and the rest (~55%) had no obvious morphological phenotype.

To test if the Δ4 allele retains any function, we injected wild-type and Δ4 mutant mRNAs at the 1 cell stage in a gain-of-function approach. Injection of 50 pg of wild-type mRNA induced phenotypes in ~80% of embryos, with over 60% displaying a severe failure to gastrulate (*Figure 1—figure supplement 3*). In contrast, injection of the same amount of mutant mRNA (and up to 200 pg per embryo) failed to induce any phenotypes, at least until 24 hpf (*Figure 1—figure supplement 3*), suggesting that the Δ4 allele is a null allele.

To understand whether the phenotypes observed in MZ*pitx2c* mutants arise from patterning defects, we analyzed the expression of several mesendodermal marker genes by whole-mount in situ hybridization. We did not observe any changes in *gsc* expression (dorsal organizer) in MZ*pitx2c* mutants (*Figure 2a*), indicating that organizer identity was induced correctly. However, the axial mesoderm marker *noto* (*flh*) was expressed in an expanded domain at mid-gastrulation in MZ*pitx2c* mutants. Analysis of *noto* expression at the one somite stage (ss) revealed that the forming notochord appeared shorter and wider (*Figure 2b,c*). Paraxial mesoderm fate was induced in MZ*pitx2c* mutants; however, *tbx16* expression domains were strongly perturbed (*Figure 2f*). Analysis of *sox17* expression, a marker of the endodermal lineage, revealed disorganized endodermal cell organization at 80% epiboly (*Figure 2g*). These data indicate that patterning and germ layer identity are established in the absence of Pitx2c function, but spatial organization is disrupted. Therefore, we hypothesize that defective cell movements during gastrulation lead to the spatial disorganization phenotypes observed in MZ*pitx2c* mutant embryos.

Pitx2 has been shown to regulate cell proliferation in other contexts (*Kioussi et al., 2002*; *Baek et al., 2003*; *Rodríguez-León et al., 2008*; *Garric et al., 2014*), therefore we assessed whether cell proliferation or cell death contributed to the phenotypes we observed. Proliferation was assessed by phospho-Histone H3 immunostaining (*Figure 2—figure supplement 1a–c*), and cell death by cleaved Caspase-3 immunostaining (*Figure 2—figure supplement 1d–f*); both analyses were performed on embryos at the 80–90% epiboly stage. No significant difference in the number of apoptotic cells was observed in MZ*pitxc2* mutants compared to wild types at late gastrulation stages. However, MZ*pitx2c* mutants had a reduced number of proliferative cells, a phenotype that may contribute to the shortened body axis observed at 24 hpf.

## MZ*pitx2c* mutants exhibit defective convergence and extension movements of mesendodermal cells during gastrulation

We next asked whether the shortened body axis phenotype observed at 24 hpf was the result of defective convergence and extension movements using different approaches. We first analyzed the expression of marker genes for the prechordal plate mesoderm (*hgg1*) and notochord (*noto*) and measured the difference between these two tissues as a read-out of axis extension (*Figure 3a–c*). At tailbud, we observed an average distance of 195.6 µm between these tissues in wild-type embryos, whereas this distance was reduced to 170.8 µm in MZ*pitx2c* mutants (n > 10, p<0.01). These data indicate that the extension of the anterior-posterior axis was less efficient in the absence of Pitx2c function.

Gene expression analysis during gastrulation revealed that the notochord appeared shorter and wider in MZ*pitx2c* mutants (*Figure 2b,c*), and so we wanted to analyze the morphology of individual notochord cells. Single blastomeres were mosaically labelled and the aspect ratio of notochord cells

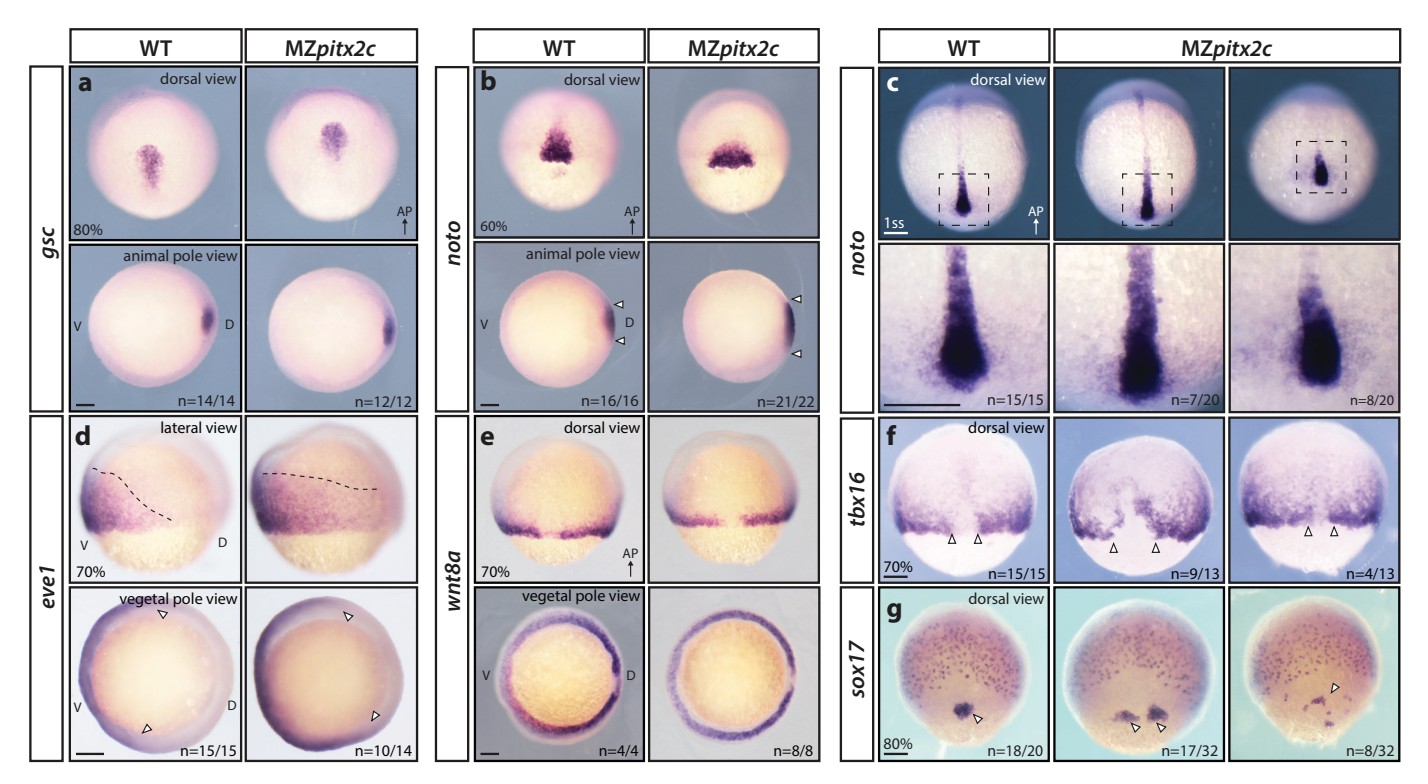

**Figure 2.** MZ*pitx2c* mutants are correctly patterned but exhibit defects in spatial organization. (a–g) Analysis of *goosecoid (gsc)* (a), *noto/flh* (b, c), *eve1* (d), *wnt8a* (e), *tbx16* (f), and *sox17* (g) expression by in situ hybridization. Expression of *gsc* (dorsal organizer) at 80% epiboly is indistinguishable between MZ*pitx2c* mutants and wild types (a). At 60% epiboly, MZ*pitx2c* mutants display an expanded domain of *noto* expression (axial mesoderm) (b) and subsequently a shorter and wider notochord at the one somite stage (ss) (c). A higher magnification of the boxed regions in panel c is shown in the lower panels. The *eve1* expression domain (ventral mesoderm) is expanded dorsally in MZ*pitx2c* mutants (d) (outlined by dashed lines and indicated by arrows in the vegetal pole views), and no obvious changes were observed in *wnt8a* expression (embryonic margin) (e). Expression of *tbx16* (paraxial mesoderm) is disrupted in the dorsal region of MZ*pitx2c* mutants (f). Expression of *sox17* (endoderm) reveals defects in dorsal forerunner cell migration (arrowheads) in 80% epiboly MZ*pitx2c* mutants (g). a-c and e-g: dorsal views, animal pole to the top; a-b: animal pole views, dorsal to the right of the image; d: lateral views, dorsal to the right of the image; d-e: vegetal pole views, dorsal to the right of the image. 'n' refers to the number of embryos with the expression pattern shown over the total number of embryos analyzed. AP, animal pole; D, dorsal; V, ventral. Scale bars, 100 μm.

DOI: https://doi.org/10.7554/eLife.34880.006

The following source data and figure supplements are available for figure 2:

**Figure supplement 1.** Proliferation is reduced in MZ*pitx2c* mutant embryos during gastrulation.

DOI: https://doi.org/10.7554/eLife.34880.007

**Figure supplement 1—source data 1.** Quantification of the number of pH3+ and cleaved Caspase-3+ cells.

DOI: https://doi.org/10.7554/eLife.34880.008

was calculated (*Figure 3d–f*). These analyses revealed that the cells in the notochord of MZ*pitx2c* mutants failed to extend mediolaterally and intercalate with neighboring cells, instead remaining more cuboidal. As notochord extension is driven by mediolateral intercalation (*Glickman et al., 2003*), we hypothesize that these defective cell shape changes and failure to intercalate with one another could contribute to the shortened and wider notochord phenotype.

To test whether the defects we observed in the axial mesoderm occurred in other migratory populations, we assessed convergence and extension movements in lateral mesendoderm by photoconversion and time-lapse imaging. Wild-type and MZ*pitx2c* mutant embryos were injected with a photoconvertible *kikGR* mRNA and raised until shield stage. Cells located 90° from the shield were photoconverted and tracked over time to assess the extent of dorsal convergence and anterior extension (*Figure 3g–j*). We observed that the photoconverted domain extended to a lesser extent in MZ*pitx2c* mutants compared to wild types (*Figure 3i*). Lateral mesendodermal cells in MZ*pitx2c* mutants also exhibited reduced convergence towards the dorsal axis compared to wild types,

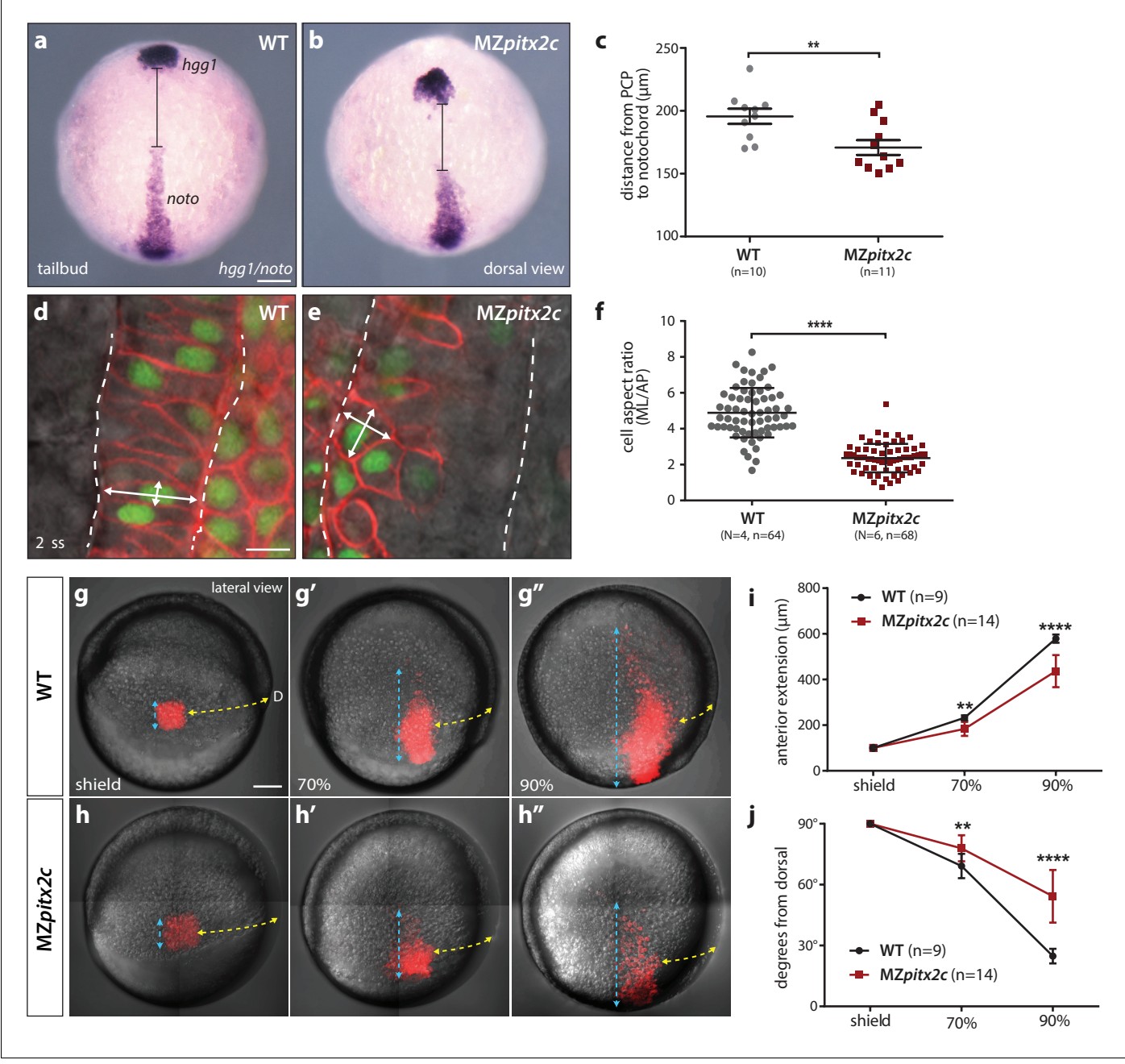

**Figure 3.** MZ*pitx2c* mutants exhibit defects in convergence and extension of mesendodermal cells. (a–c) In situ hybridization for *hgg1* (prechordal plate) and *noto/flh* (notochord) expression at tailbud. Quantification of the distance between the *hgg1* and *noto* domains is shown in (c); the distance measured is indicated by the lines shown in panels a and b. (d–f) Analysis of cell shapes in the notochord by mosaically labelling cells with membrane tdTomato and nuclear H2B-GFP. Quantification of the ratio between the mediolateral (ML) width and anterior-posterior (AP) length of cells in the notochord is shown in (f); n > 60 cells from at least four embryos. MZ*pitx2c* mutants exhibit a reduced distance between the prechordal plate and notochord (b, c), and axial mesodermal cells of the notochord fail to intercalate and elongate mediolaterally (e, f), indicative of defects in mesodermal convergence and axis extension. (g–i) Analysis of lateral mesendodermal migration. Embryos were injected with photoconvertible *kikGR* mRNA at the 1 cell stage; a 100 μm² region of mesendodermal cells located 90° from the dorsal organizer was photoconverted at shield stage and the photoconverted population was tracked until 90% epiboly (g, h). To quantify the extent of anterior migration, the length of the photoconverted population was measured (shown by the blue dashed lines) (i). Convergence towards the dorsal midline (shown by the yellow dashed lines) was assessed by measuring the circumferential angle of the location of the dorsal-most photoconverted cells from the dorsal midline (j). MZ*pitx2c* mutants exhibit reduced dorsal convergence as well as reduced anterior extension of lateral mesendodermal cells during late gastrulation when compared to wild types (i, j). a-b, d-e: dorsal views, animal pole to the top; g-h: lateral views, dorsal to the right of the image. D, dorsal. **p<0.01, ****p<0.0001 by unpaired *t*-test. Scale bars, 100 μm (a, g) or 10 μm (d).

*Figure 3 continued on next page*

*Figure 3 continued*

DOI: https://doi.org/10.7554/eLife.34880.009

The following source data is available for figure 3:

**Source data 1.** Quantification of the distance between *hgg1* and *noto* expression domains.
DOI: https://doi.org/10.7554/eLife.34880.010
**Source data 2.** Quantification of the aspect ratio of mosaically labelled cells in the notochord of WT and MZ*pitx2c* mutant embryos.
DOI: https://doi.org/10.7554/eLife.34880.011
**Source data 3.** Quantification of anterior extension and dorsal convergence of photoconverted cells in WT and MZ*pitx2c* mutant embryos.
DOI: https://doi.org/10.7554/eLife.34880.012

altogether suggesting that the gastrulation movements of dorsal convergence and anterior extension are both compromised in MZ*pitx2c* mutants.

We also tested whether endodermal cell migration was affected in MZ*pitx2c* mutants. Endodermal cells undergo two phases of migration during gastrulation: a random walk motion that is thought to be an efficient way to populate the embryo during early gastrulation, followed by oriented, persistent migration that drives endodermal convergence to the midline at late gastrulation (*Concha and Adams, 1998*; *Pézeron et al., 2008*; *Woo et al., 2012*). Some of the inductive cues underlying these behaviors have been elucidated, including the dependence of these movements on Nodal signaling (*Woo et al., 2012*), an upstream regulator of *pitx2* expression. Therefore, we decided to analyze these behaviors in MZ*pitx2c* mutants using the *Tg(sox17:GFP)* line to visualize the endodermal cells. As previously reported, we observed the random walk motion of endodermal cells during early gastrulation (6–7 hpf), with little directionality and minimal track displacement (~35 μm) (*Figure 4a,c–e*). We observed that endodermal cells in MZ*pitx2c* mutants behaved similarly as those in wild-type embryos at this stage, albeit with a slightly reduced speed (*Figure 4b–e*). At late gastrulation, we observed that the migration behavior in wild-type embryos became more directed and persistent, with increased speed (*Figure 4f,h–j*). Strikingly, this transition was abrogated in MZ*pitx2c* mutants (*Figure 4g–i*). Endodermal cells in MZ*pitx2c* mutants displayed meandering trajectories rather than oriented migration, as evidenced by decreased overall displacement (*Figure 4i*) and a lower straightness index (*Figure 4j*), and endodermal cell speed remained the same as during early gastrulation (*Figure 4h*). We then examined the anterior endoderm at 24 hpf in the *Tg(sox17: GFP)* background. Using the level of the fifth pharyngeal pouch as a landmark to measure endodermal width, we observed that it was significantly increased in MZ*pitx2c* mutants compared to wild types (*Figure 4m*). Taken together, these data indicate that Pitx2c is required for the transition from random walk to oriented, persistent endodermal cell migration at late gastrulation stages. We hypothesize that these aberrant behaviors during gastrulation lead to the increased width of the endodermal sheet.

## Pitx2c promotes oriented cell divisions during gastrulation

Oriented cell division contributes to tissue elongation and shaping in different developmental contexts and has been observed during zebrafish epiboly (*Gong et al., 2004*); therefore, we assessed the orientation of cell divisions during gastrulation in wild types and MZ*pitx2c* mutants. To visualize dividing cells, we injected membrane *tdTomato* mRNA and nuclear *H2B-GFP* mRNA at the one-cell stage. Live-imaging was performed between 60–90% epiboly and epiblast cell divisions were assessed in a region located 90° from the dorsal organizer (*Figure 5a*). Similar to what has been previously reported (*Concha and Adams, 1998*; *Gong et al., 2004*), we observed that cell divisions occur preferentially along the animal-vegetal axis, with the majority of cell division angles greater than 45° from the margin (*Figure 5b,d*). In contrast we observed that in MZ*pitx2c* mutants, cell divisions occurred without directional bias (*Figure 5c,e*). We then tested whether wild-type *pitx2c* mRNA could rescue this phenotype, and indeed observed a wild-type-like pattern of cell division angles (*Figure 5g*), indicating that global *pitx2c* overexpression can rescue the cell division phenotype in MZ*pitx2c* mutant embryos.

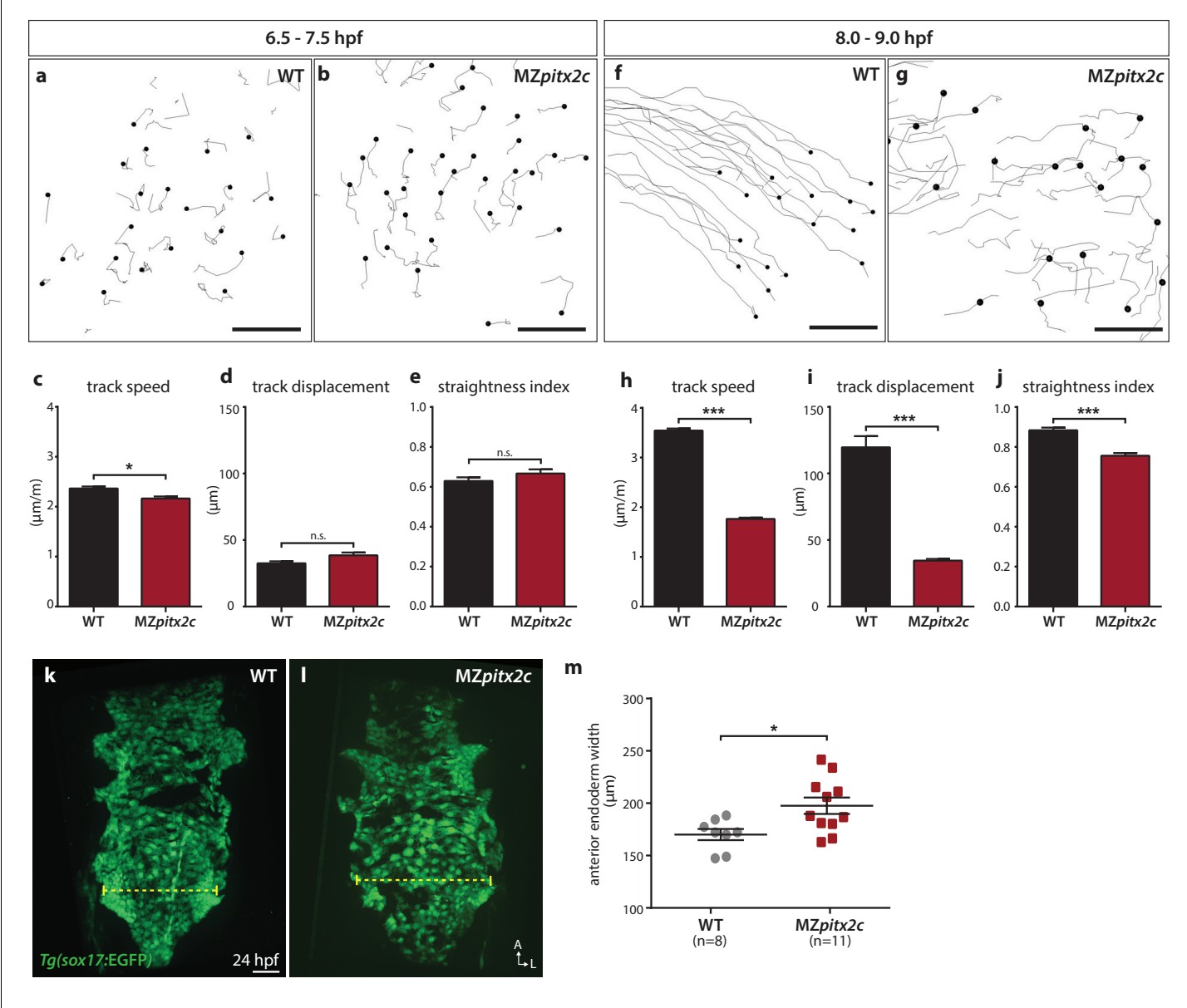

**Figure 4.** Pitx2c promotes oriented, persistent endodermal cell migration. (a–e) Time-lapse imaging was performed to assess the early migration (random walk pattern from 6.5 to 7.5 hpf) and (f–j) late migration (oriented and persistent pattern from 8 to 9 hpf) of endodermal cells labelled by *Tg (sox17*:GFP) expression. Representative tracks of endodermal cells are shown from 6.5 to 7.5 hpf (a, b) and from 8 to 9 hpf (f, g) from wild-type and MZ*pitx2c* mutant embryos. Quantification of speed (c, h), track displacement (d, i), and track straightness (e, j) from wild-type and MZ*pitx2c* mutant embryos are shown in the graphs below (n > 3 embryos, 20–40 tracks per embryo). While early random walk motions are similar to wild types (a–e), later endodermal cell persistence and directional migration are significantly affected in MZ*pitx2c* mutants (f–j). (k–m) Analysis of anterior endodermal derivatives at 24 hpf in wild-type (k) and MZ*pitx2c* mutant (l) embryos. The width of the endoderm (shown by the yellow dashed lines) was measured at the level of the fifth pharyngeal pouch and quantified from at least eight embryos (m). Increased endoderm width is observed in MZ*pitx2c* mutants compared to wild types. *p<0.05, ***p<0.001 by unpaired *t*-test. Scale bars, 40 µm.

DOI: https://doi.org/10.7554/eLife.34880.013

The following source data is available for figure 4:

**Source data 1.** Migration parameters of endodermal cells in WT and MZ*pitx2c* mutant embryos.
DOI: https://doi.org/10.7554/eLife.34880.014

**Source data 2.** Anterior endoderm width in WT and MZ*pitx2c* mutant embryos at 24 hpf.
DOI: https://doi.org/10.7554/eLife.34880.015

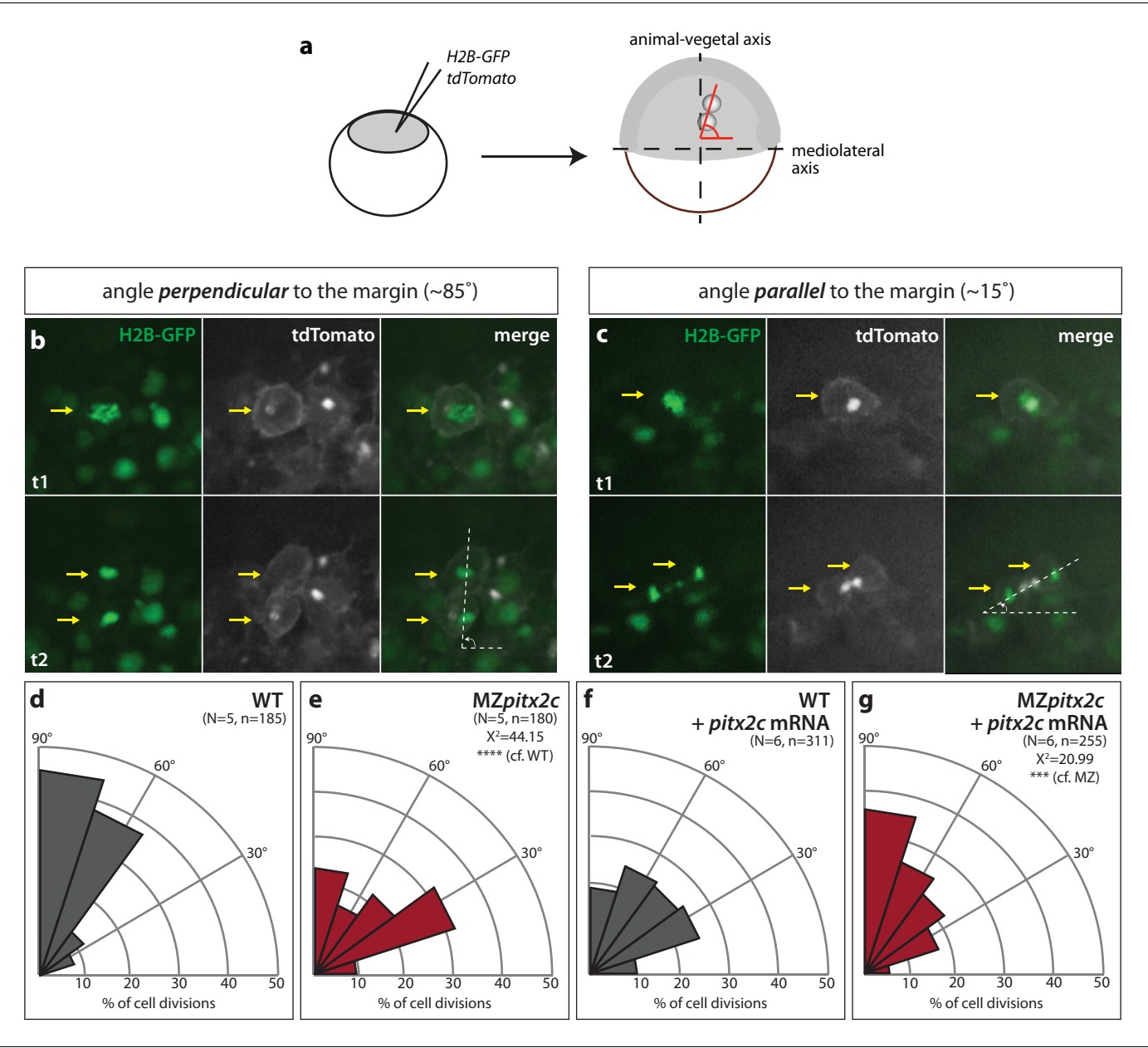

**Figure 5.** Modulation of *pitx2c* expression affects cell division orientation during gastrulation. (**a**) Schematic of the experimental design to assess the orientation of cell division in the embryo. Embryos were injected at the 1 cell stage with *H2B-GFP* and membrane *tdTomato* mRNA. Time-lapse imaging was performed from shield stage for 90 mins; quantification was performed on at least five embryos per condition with 30–60 cell divisions counted per embryo. (**b**) Example of a cell division oriented perpendicular to the mediolateral (ML) axis shown at two consecutive time points, t1 (upper panel) and t2 (lower panel). (**c**) Example of a cell division oriented parallel to the ML axis shown at two consecutive time points, t1 and t2. Yellow arrows point to the particular cell division that is measured by the white dashed lines. All images are oriented with dorsal to the right, animal pole to the top. (**d–g**) Rose diagrams indicating the orientation of cell division axes in wild types (**d**) and MZ*pitx2c* mutants (**e**), and after injection of *pitx2c* mRNA at the 1 cell stage in wild types (**f**) and MZ*pitx2c* mutants (**g**). The proportion of cell division angles is indicated relative to the ML axis. MZ*pitx2c* mutants exhibit a higher proportion of cell divisions oriented less than 45° from the ML axis (**e**) as compared to wild-type embryos (**d**). Injection of 25 pg of *pitx2c* mRNA at the 1 cell stage randomizes cell division orientation in wild-type embryos (**f**), whereas it increases the proportion of cell divisions occurring perpendicular to the ML axis in MZ*pitx2c* mutant embryos (**g**). Chi-square tests were performed on binned data of cell division angles.
DOI: https://doi.org/10.7554/eLife.34880.016

The following source data is available for figure 5:

**Source data 1.** Binned angles of cell division.
DOI: https://doi.org/10.7554/eLife.34880.017

# Pitx2c promotes convergence and extension movements during gastrulation in a cell non-autonomous manner

To address the autonomy of Pitx2c function during convergence and extension movements, we performed different transplantation experiments. We first transplanted two populations of cells into wild-type hosts at 30% epiboly: cells from wild-type donors injected with membrane *tdTomato* mRNA and cells from MZ*pitx2c* mutant donors injected with *LIFEACT-GFP* mRNA (*Figure 6a*). Donor embryos were co-injected with 100 pg *ndr2* mRNA to drive transplanted cells towards a mesendodermal fate (*Krieg et al., 2008*). We then measured the distance that wild-type and MZ*pitx2c* mutant donor cells migrated along the anterior-posterior axis (*Figure 6—figure supplement 1a,d*). Although there was a tendency for wild-type donor cells to migrate slightly further along the axis than MZ*pitx2c* mutant donor cells, we did not observe any statistically significant difference (*Figure 6—figure supplement 1d*). Moreover, we observed that both populations of transplanted cells mixed with one another (*Figure 6—figure supplement 1a'*), arguing against differential adhesion properties between the two populations of cells.

To test the hypothesis that Pitx2c functions cell non-autonomously, we transplanted wild-type cells labelled with FITC dextran into wild-type or MZ*pitx2c* mutant hosts and analyzed the transplants at ~60% epiboly (*Figure 6—figure supplement 1b,c,e*). To quantify the spread of transplanted cells, we overlaid images of embryos with a 35 × 35 grid and quantified the proportion of squares containing at least one transplanted cell. We observed a significant reduction in the spread of wild-type cells transplanted into MZ*pitx2c* mutant hosts compared to wild-type hosts (*Figure 6—figure supplement 1e*), suggesting that the MZ*pitx2c* mutant host environment affects the migratory capability of the transplanted wild-type cells. We also raised the transplanted embryos until two ss and analyzed how far the transplanted cells had migrated along the anterior-posterior axis (*Figure 6—figure supplement 1b', c'*). Transplanted cells in wild-type hosts were detected along the entire length of the axis (*Figure 6b'*); however, wild-type cells transplanted into MZ*pitx2c* mutant hosts extended ~18% less along the AP axis (*Figure 6—figure supplement 1c,f*). Taken together, these data suggest that Pitx2c functions cell non-autonomously to regulate cell migration during gastrulation.

In order to evaluate the migratory behavior of transplanted cells at higher resolution, we performed live imaging of transplanted cell dynamics in our different transplantation scenarios. Images were acquired every 90 s for 30–60 min and cell morphology parameters and protrusive activity

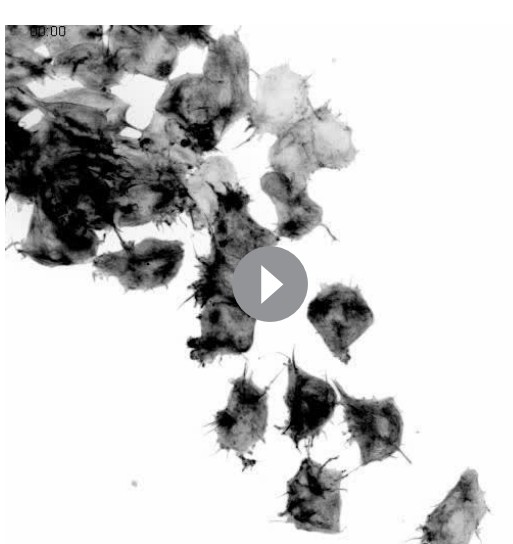

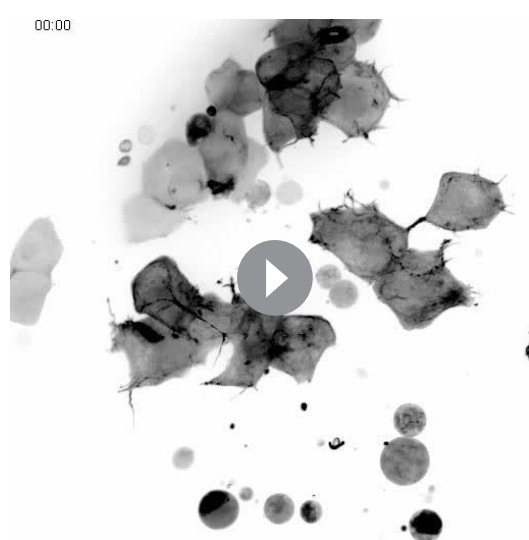

**Video 1.** Cells from a wild-type donor labelled with *LIFEACT-GFP* mRNA transplanted into a wild-type host and imaged at late gastrulation stages (~80% epiboly).
DOI: https://doi.org/10.7554/eLife.34880.025

**Video 2.** Cells from a wild-type donor labelled with *LIFEACT-GFP* mRNA transplanted into a MZ*pitx2c* mutant host and imaged at late gastrulation stages (~80% epiboly).
DOI: https://doi.org/10.7554/eLife.34880.026

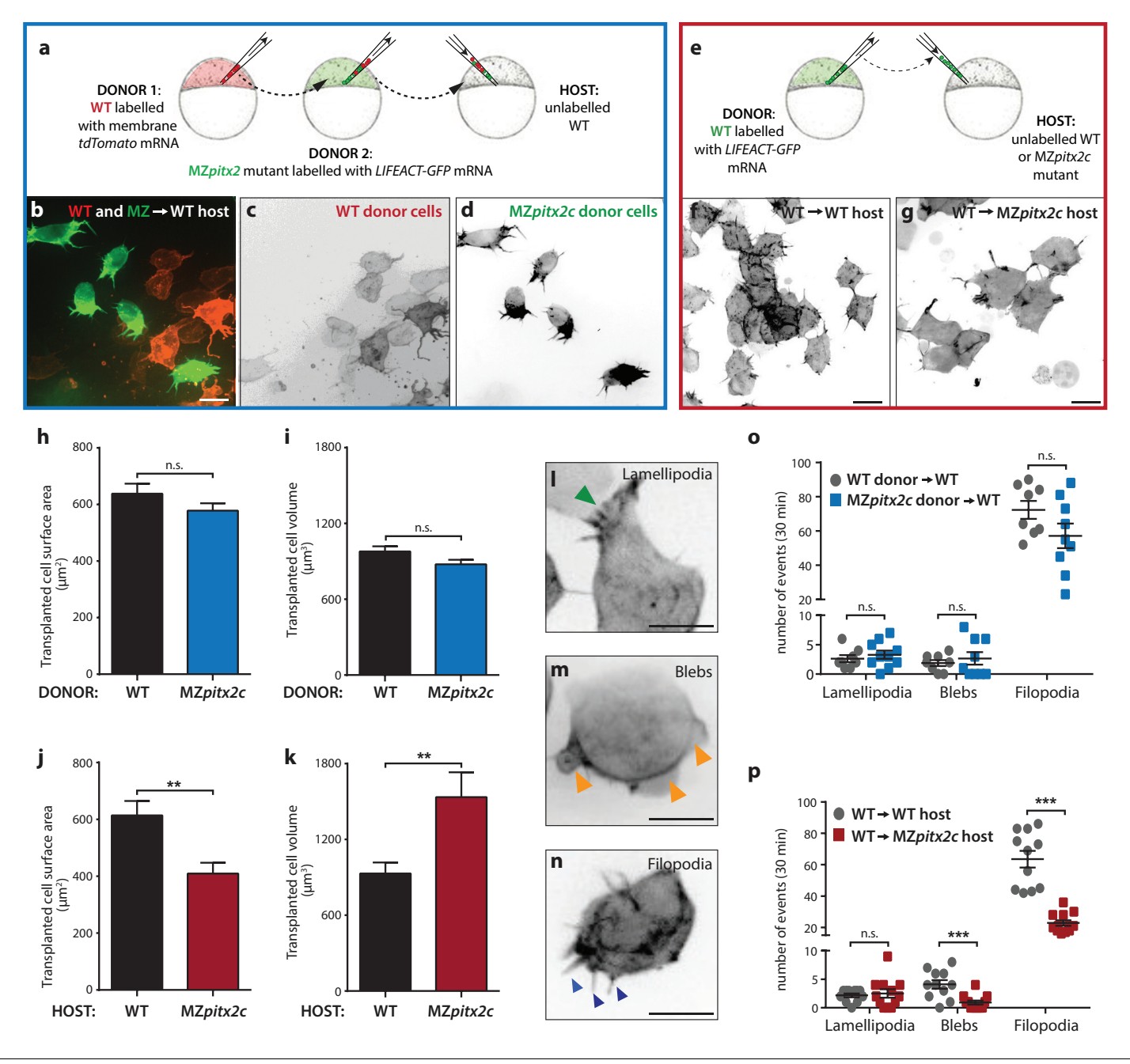

**Figure 6.** Pitx2c functions cell non-autonomously to promote convergence and extension during gastrulation. (a–d) Wild-type donor cells labelled with membrane *tdTomato* mRNA and MZ*pitx2c* donor cells labelled with *LIFEACT-GFP* mRNA were transplanted into wild-type host embryos. Live imaging of successfully transplanted hosts was performed from 70–90% epiboly. (e–g) Wild-type donor cells labelled with *LIFEACT-GFP* mRNA were transplanted into wild-type (f) or MZ*pitx2c* mutant (g) host embryos. (h–k) Analysis of transplanted cell surface area and cell volume in the different transplantation approaches shown in panel a (h, i) and panel e (j, k). No differences in cell surface area or volume were observed between wild-type and MZ*pitx2c* mutant donor cells transplanted into a wild-type host (h, i). In contrast, wild-type cells transplanted into a MZ*pitx2c* mutant host appeared flattened, with a larger surface area and reduced volume, as compared to cells transplanted into a wild-type host (j, k). (l–p) Quantification of protrusions during mesendodermal migration in the different transplantation approaches shown in panel a (o) and panel e (p). Examples of lamellipodia (l; green arrowheads), blebs (m; orange arrowheads), and filopodia (n; blue arrowheads) formed during mesendodermal migration. The number of each type of protrusions observed over a 30 min period was quantified at late gastrulation (70–90% epiboly) (n > 10 cells from at least three embryos). Wild-type cells transplanted into mutant hosts exhibit less protrusive behavior (p) than wild-type cells transplanted into wild-type hosts. **p<0.01, ***p<0.001 by unpaired *t*-test. Scale bars, 10 μm.

DOI: https://doi.org/10.7554/eLife.34880.018

*Figure 6 continued on next page*

*Figure 6 continued*

The following source data and figure supplements are available for figure 6:

**Source data 1.** Quantification of transplanted cell surface area and volume.
DOI: https://doi.org/10.7554/eLife.34880.023
**Source data 2.** Quantification of the number of protrusive events.
DOI: https://doi.org/10.7554/eLife.34880.024
**Figure supplement 1.** Analysis of cell autonomy.
DOI: https://doi.org/10.7554/eLife.34880.019
**Figure supplement 1—source data 1.** Quantification of the angle of extension of transplanted WT and MZpitx2c cells.
DOI: https://doi.org/10.7554/eLife.34880.020
**Figure supplement 1—source data 2.** Quantification of transplanted cell dispersal.
DOI: https://doi.org/10.7554/eLife.34880.021
**Figure supplement 1—source data 3.** Quantfication of the angle of extension of transplanted cells in WT and MZpitx2c hosts at 2 ss.
DOI: https://doi.org/10.7554/eLife.34880.022

were quantified. During mesendodermal migration, three types of protrusions are observed: filopodia, lamellipodia, and blebs (*Diz-Muñoz et al., 2010*) (*Figure 6l–n*). These protrusive behaviors are thought to provide specific functions during directed migration, and so we quantified the number of each protrusive event observed over a 30 min period during late gastrulation (70–90% epiboly). We transplanted a mixed population of donor cells (i.e. from wild-type and MZ*pitx2c* mutant embryos) (*Figure 6a–d*) and did not observe any significant differences in terms of transplanted cell surface area or volume (*Figure 6h,i*). Moreover, we did not observe any differences in the number of lamellipodia, blebs, or filopodia when comparing wild-type and MZ*pitx2c* mutant cells transplanted into a wild-type host (*Figure 6o*).

We then performed the same analysis on wild-type cells injected with *LIFEACT-GFP* mRNA and transplanted into wild-type or MZ*pitx2c* mutant hosts (*Figure 6e*). From time-lapse imaging, wild-type cells in MZ*pitx2c* mutant hosts appeared larger than wild-type cells in wild-type hosts (*Figure 6f and g*, *Videos 1* and *2*). Indeed, quantification showed that wild-type cells transplanted into MZ*pitx2c* mutant hosts exhibited increased cell surface area and decreased cell volume compared to wild-type cells transplanted into wild-type hosts. Live imaging also suggested that wild-type cells transplanted into MZ*pitx2c* mutant hosts exhibited fewer and more stable protrusions (*Video 2*). We indeed found that the number of filopodia and blebs was reduced in wild-type cells transplanted into MZ*pitx2c* mutant hosts (*Figure 6p*), suggesting that inhibition of cell migration may arise from defective interactions between the cells and underlying matrix in the absence of Pitx2c function.

## Transcriptomic and candidate gene expression approaches identify potential pathways downstream of Pitx2c

To screen for genes potentially acting downstream of Pitx2c, we used both transcriptional profiling and a candidate gene approach. We performed microarray analysis on MZ*pitx2c* mutant embryos at shield stage, as well as on dome stage embryos following injection of 25 pg *pitx2c* mRNA as a gain-of-function (GOF) approach (*Figure 1—figure supplement 3*), and both sets of embryos were compared to stage-matched controls. We performed gene enrichment and functional annotation analyses on the transcriptomic datasets using the Database for Annotation, Visualization and Integrated Discovery (DAVID) (*Huang et al., 2009a*; *2009b*) to gain an overview of the biological processes, molecular functions, and KEGG pathways overrepresented in the datasets (*Figure 7—figure supplement 1*).

To further test whether early embryonic patterning was affected in MZ*pitx2c* mutants, we examined the expression levels of several key mesendodermal specification and dorsal organizer genes (*Figure 7—figure supplement 2a*). We observed that some genes in these categories were differentially expressed in the *pitx2c* GOF transcriptome, including *chd*, *foxa2*, *sox32*, *pcdh8*, and *vox*, whereas the expression level of many of the genes in these categories were not highly changed following the loss of Pitx2c function (*Figure 7—figure supplement 2a*).

From the list of differentially expressed genes, we selected several candidates for further analysis. One of the most highly upregulated genes in the GOF dataset was *cxcl12b,* which encodes the chemokine Cxcl12b that has previously been implicated in endodermal migration (*Mizoguchi et al., 2008*; *Nair and Schilling, 2008*). *cxcl12b* mRNAs are maternally contributed and subsequently found in the mesodermal lineage during gastrulation; the gene encoding its receptor Cxcr4a is expressed by endodermal cells (*Mizoguchi et al., 2008*; *Nair and Schilling, 2008*). We analyzed mRNA levels of *cxcl12b* and *cxcr4a* in wild-type, MZ*pitx2c* mutant embryos, and *pitx2c* GOF embryos at 80% epiboly. These data indicate that *cxcl12b* is positively regulated by Pitx2c, as mRNA levels were reduced by half in MZ*pitx2c* mutants and increased ~5 fold after *pitx2c* overexpression (*Figure 7a*), confirming the observed increase in the microarray analysis. We did not observe any appreciable changes in *cxcr4a* mRNA levels (*Figure 7b*) after manipulating *pitx2c* expression.

It has been shown that Cxcl12b-Cxcr4a signaling controls ECM-integrin-dependent adhesion during gastrulation cell movements (*Nair and Schilling, 2008*). Therefore, we considered whether genes in this pathway were modulated in the transcriptomics datasets. We observed that several genes encoding integrin subunits and ECM remodelers were affected (*Figure 7—figure supplement 2b*), while the expression level of a few genes encoding cadherins were also changed (*Figure 7—figure supplement 2c*). Using a candidate gene approach, we investigated whether expression of *itgb1b* was affected as it is one of the most highly expressed integrin genes during gastrulation (*Nair and Schilling, 2008*). We found that *itgb1b* mRNA levels were increased ~3 fold in MZ*pitx2c* mutants at 80% epiboly and observed a slight reduction in mRNA levels following *pitx2c* GOF (*Figure 7c*). We also sought to determine whether fibronectin deposition was affected given changes to the upstream chemokine signaling and integrin levels, and because *fn1a* is highly expressed by mesodermal cells during gastrulation (*Trinh and Stainier, 2004*). At two ss, we could observe fibrillar fibronectin structures in the ECM surrounding the endoderm (*Figure 7d–f*). In contrast, a less dense fibrillar network was observed in MZ*pitx2c* mutants (*Figure 7g–i*), which we hypothesize could impact both cell shape and migratory capabilities in a cell non-autonomous manner.

The transcriptomic analyses also identified two genes that function within the planar cell polarity (PCP) signaling pathway: *frzb*, which encodes a secreted frizzled related protein, and *wdpcp*, which encodes a WD repeat containing PCP effector protein. We analyzed mRNA levels by qPCR in MZ*pitx2c* mutant embryos and following *pitx2c* GOF at 80% epiboly (*Figure 7—figure supplement 3a,b*). We confirmed that *wdpcp* mRNA expression was reduced in MZ*pitx2c* mutants but did not see strong upregulation following *pitx2c* GOF. Additionally, we confirmed that *frzb* mRNA expression was significantly upregulated following *pitx2c* misexpression but did not observe any significant changes in MZ*pitx2c* mutants. We also took a candidate approach to examine mRNA levels of components of the PCP pathway, as several aspects of the MZ*pitx2c* mutant phenotype resemble zebrafish PCP mutants, including the shortened, wider body axis (*Marlow et al., 1998*; *Sepich et al., 2000*; *Topczewski et al., 2001*; *Jessen et al., 2002*; *Kilian et al., 2003*; *Yin et al., 2008*). We analyzed mRNA levels of genes encoding PCP components at 80% in *pitx2c* gain- and loss-of-function embryos. No changes in *prickle1a* or *gpc4* mRNA expression levels were detected (data not shown). We observed that *vangl2* mRNA expression levels were reduced in MZ*pitx2c* mutants compared to wild types and were slightly upregulated in *pitx2c* GOF embryos (*Figure 7—figure supplement 3c*). We also observed a significant reduction of *wnt5b* mRNA expression in MZ*pitx2c* mutant embryos (*Figure 7—figure supplement 3d*), and although *pitx2c* GOF could induce *wnt11* expression (*Figure 7—figure supplement 3e*), no changes in expression were detected in MZ*pitx2c* mutants. These data suggest that Pitx2c regulates the expression of a subset of PCP components, and transcriptional changes to these genes could potentially also contribute to the phenotypes observed in MZ*pitx2c* mutants.

## Discussion

Cell migration during gastrulation shapes the embryo and positions cells in their correct locations to receive signaling and patterning information. A precise coordination of cell intrinsic processes, including polarized protrusive behavior and dynamic actin remodeling, with extrinsic processes such as chemokine signaling and ECM assembly is required for cell migration. As such, a cell's migratory behavior is dependent on its ability to respond to migratory cues as well as its environment, and

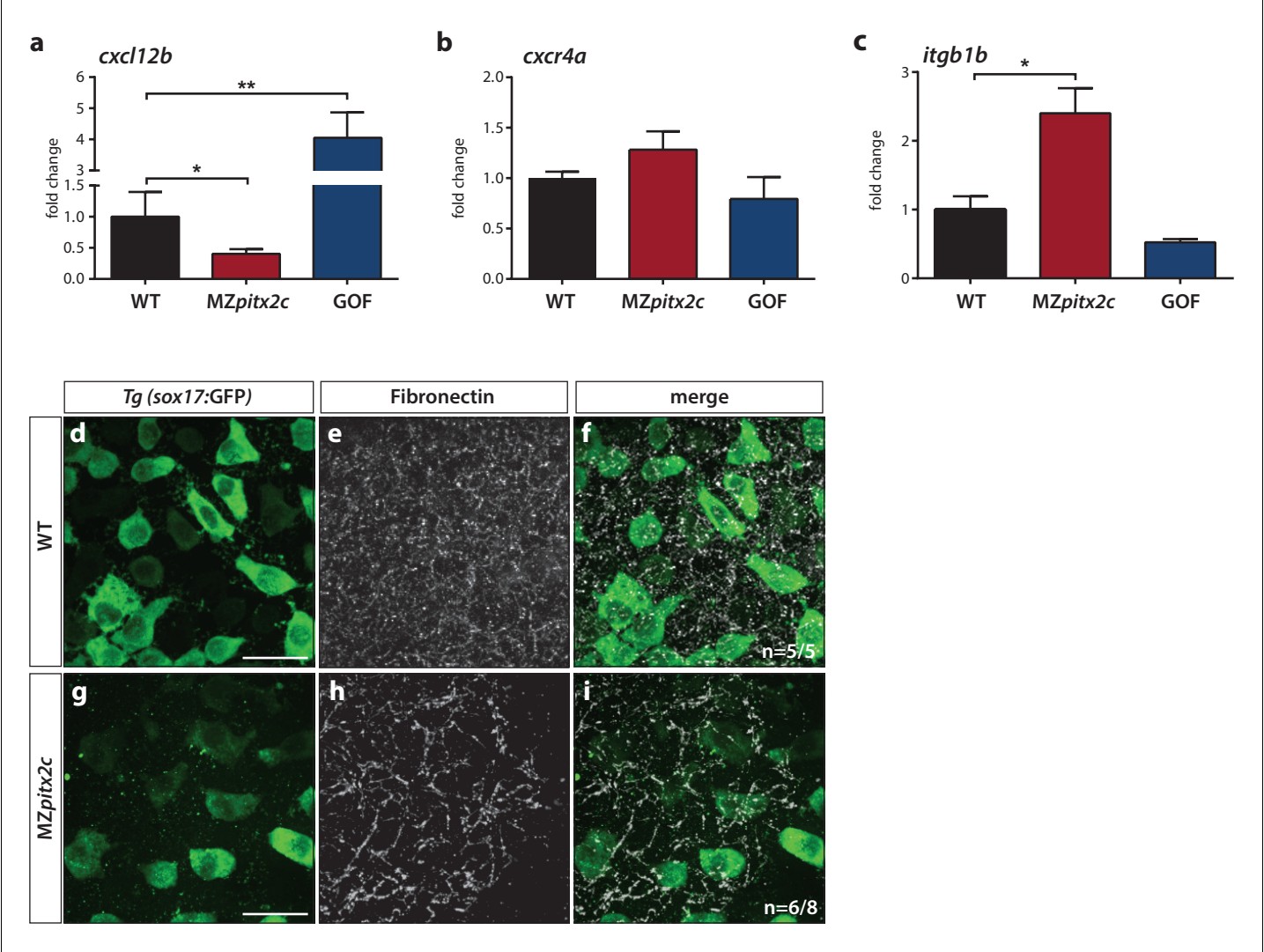

**Figure 7.** Pitx2c modulates the chemokine and ECM-integrin signaling axis. Analysis of the chemokine signaling axis and ECM factors that control endodermal migration. (a–c) qPCR analysis of *cxcl12b* (a), *cxcr4a* (b), and *itgb1b* (c) mRNA levels in wild-type, MZ*pitx2c* mutant, and *pitx2c* GOF embryos at 80% epiboly. (d–i) Fibronectin staining in wild-type (d–f) and MZ*pitx2c* mutant (g–i) embryos in the *Tg(sox17:GFP)* background to visualize endodermal cells. While fibronectin deposition appears fibrillar by the two ss in wild-type embryos (e), a less dense network is observed in MZ*pitx2c* mutants (h). 'n' refers to the number of embryos with the expression pattern shown over the total number of embryos analyzed. *p<0.05, **p<0.01 by unpaired *t*-test. Graphs indicate the average fold change ± SEM from at least three biological replicates analyzed in duplicates. Scale bars, 20 µm.
DOI: https://doi.org/10.7554/eLife.34880.027

The following figure supplements are available for figure 7:

**Figure supplement 1.** Gene enrichment analysis of differentially expressed genes following *pitx2c* gain- (GOF) or loss- (MZ*pitx2c*) of-function.
DOI: https://doi.org/10.7554/eLife.34880.028

**Figure supplement 2.** Profiling of patterning, adhesion, and ECM genes in *pitx2c* GOF and LOF transcriptomes.
DOI: https://doi.org/10.7554/eLife.34880.029

**Figure supplement 3.** Expression analysis of genes encoding PCP components.
DOI: https://doi.org/10.7554/eLife.34880.030

crosstalk between intrinsic and extrinsic processes is important to modulate their contributions during different phases of migration.

Our data reveal that Pitx2 plays an important role during gastrulation movements. In this manuscript, we show that the loss of maternal and zygotic Pitx2c function leads to reduced axis extension. We observed that MZ*pitx2c* mutant embryos exhibit defects in convergence and extension

movements in their axial and lateral mesoderm, and that persistent, oriented endodermal migration is impaired at late gastrulation stages. Using transplantation experiments, we observed that Pitx2c functions cell non-autonomously during migration to promote the protrusive activity of mesendodermal cells. We also identified differential expression of genes known to affect cell behaviors during gastrulation. Based on these data, we propose a model whereby Pitx2 promotes chemokine signaling and integrin-ECM organization to promote mesendodermal cell migration, which itself is required for axis extension (*Figure 8*).

## An early wave of Pitx2c expression is evolutionarily conserved

Pitx2 is known for its role in many tissues and developmental contexts, including during the establishment of left-right asymmetry, skeletal muscle patterning, ocular development, and cardiac development, thus reflecting its dynamic function during embryonic and adult life. Here, we discuss an early role for Pitx2 during gastrulation. We and others (*Essner et al., 2000*; *Faucourt et al., 2001*) report *pitx2c* expression at the margin of the zebrafish embryo at the onset of gastrulation. Interestingly, this early wave of expression is not unique to zebrafish. Recently, *pitx2* expression has been reported in the early avian (*Torlopp et al., 2014*) and rabbit (*Plöger and Viebahn, 2018*) embryos preceding the formation of the primitive streak, as well as in the dorsal lip of the early frog gastrula (*Blitz et al., 2017*; *Ding et al., 2017*). In mouse, *Pitx2* expression is detected at embryonic day (E) 6.25 by transcriptome profiling (*Mitiku and Baker, 2007*). Single cell RNAseq datasets indicate that *Pitx2* expression can be detected during gastrulation from E6.5 to E7.5 in the epiblast, pharyngeal mesoderm, nascent mesoderm, posterior mesoderm, allantois, and visceral endoderm (data obtained from online resource found at http://gastrulation.stemcells.cam.ac.uk). Together, these observations indicate that the early wave of *pitx2* expression is highly evolutionarily conserved.

*Pitx2* was recently described as one of the earliest markers of embryonic anterior-posterior polarity and as essential for axis formation (*Torlopp et al., 2014*). This function appears to have been retained during mammalian evolution as a similar relationship was recently reported in rabbit embryos (*Plöger and Viebahn, 2018*); however, little is known about the role of Pitx2 at these early stages in mice. In zebrafish, previous reports suggest that Pitx2c can affect germ layer specification (*Faucourt et al., 2001*). In agreement, we observe that misexpression of *pitx2c* mRNA at the 1 cell

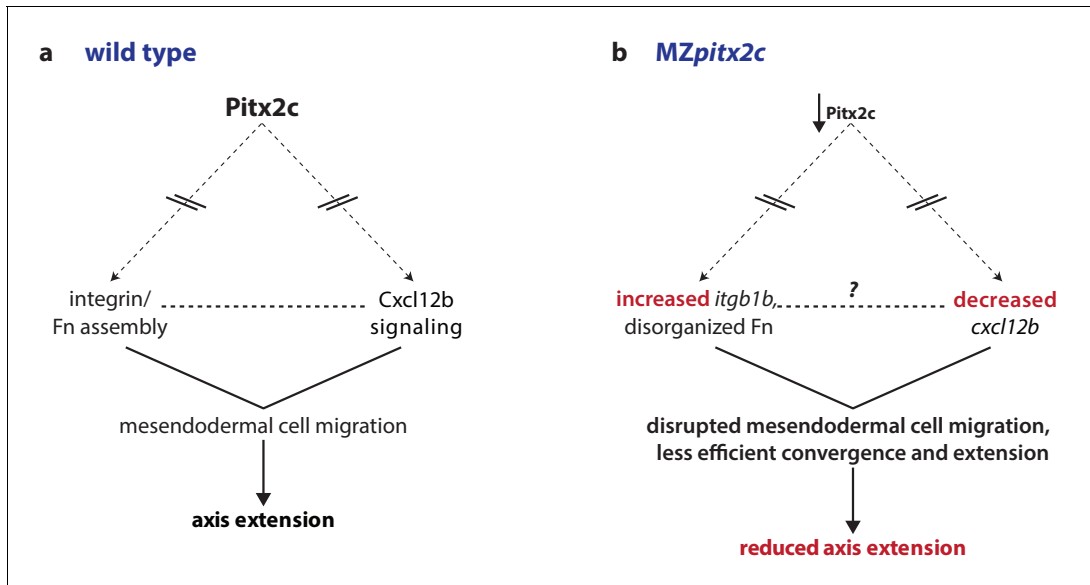

**Figure 8.** Proposed model of Pitx2c function during gastrulation. (**a**) Altogether, these data suggest a model whereby Pitx2c acts during late gastrulation to affect chemokine signaling (via *cxcl12b*) and integrin-Fn assembly (via *itgb1b*). These factors cooperate to drive mesendodermal cell migration which contributes to elongation of the embryo along the anterior-posterior axis. (**b**) In MZ*pitx2c* mutants, we observed increased *itgb1b* mRNA levels and decreased *cxcl12b* mRNA levels. We hypothesize that these transcriptional changes lead to disrupted mesendodermal cell migration and less efficient convergence and extension movements during gastrulation, ultimately leading to reduced anterior-posterior axis elongation.
DOI: https://doi.org/10.7554/eLife.34880.031

stage induces the expression of many patterning genes, including *dkk1a* and *chd* (**Figure 7—figure supplement 2**). However, the expression levels of the vast majority of genes appear unchanged at shield stage in MZ*pitx2c* mutants, including genes encoding key transcription factors that regulate germ layer specification, including Gata5 and Sox32, and signaling molecules such as Ndr1 and Ndr2 (**Figure 7—figure supplement 2**).

It is interesting to consider whether the compensatory response to genetic perturbation of *Pitx2* is also conserved between species. During early chick development, knockdown of Pitx2 is followed by upregulation of a related gene, *Pitx1*, that can compensate for the loss of Pitx2 (**Torlopp et al., 2014**). In zebrafish, mRNA levels of *pitx1* and *pitx3* are very low in wild-type embryos at late gastrulation. However, levels of both transcripts are increased 5 to 10-fold in MZ*pitx2c* mutants at 80% epiboly (**Figure 1—figure supplement 2**), suggesting that a similar response occurs in fish. We thus hypothesize that upregulation of Pitx1 or Pitx3 partially compensates for the loss of Pitx2c function.

## Pitx2 promotes mesendodermal cell migration during late gastrulation

From the analysis of endodermal cell migration in MZ*pitx2c* mutants, we observed that Pitx2 promotes the transition from random walk to oriented, persistent migration. It has been proposed that Cxcl12-Cxcr4-dependent expression of *itgb1b* tethers the endoderm and mesoderm to promote endodermal migration towards the anterior pole (**Nair and Schilling, 2008**); another study suggests that these cues act as a chemotactic signal to promote the formation and orientation of filopodia in endodermal cells (**Mizoguchi et al., 2008**). In our analyses, we also observed transcriptional changes in these genes and in fibronectin organization in MZ*pitx2c* mutant embryos, as well as disrupted cytoskeletal dynamics, thus supporting these models. We speculate that modified interactions between the focal adhesions and the underlying ECM could cause the cell non-autonomous migration defects we observed. Moreover, morpholino knockdown of *cxcr4a* or *cxcl12b* delays endodermal migration, and also reduces the number of filopodia in endodermal cells (**Mizoguchi et al., 2008**). Therefore, we hypothesize that the lack of a chemokine signal also impacts cytoskeletal dynamics, as observed in our cell transplantation experiments.

Oriented cell division along the animal-vegetal axis has been observed during zebrafish epiboly (**Gong et al., 2004**), but whether it is able to drive axis extension or morphogenesis at these stages remains unclear. Here we show that in the absence of Pitx2c function, the direction of mitoses of deep cells becomes randomized during epiboly. However, our data do not resolve whether this defect directly leads to the axis extension defects. It is important to note, though, that the hypothesis that Pitx2 regulates oriented cell division has also been proposed from observing outflow tract development in the mouse *Pitx2* null model (**Meilhac et al., 2004**; **Ai et al., 2006**; **Ma et al., 2013**) and during asymmetric development of the ovary in the chick (**Rodríguez-León et al., 2008**).

## A cell non-autonomous role for Pitx2c during mesendodermal cell migration

Transcriptomic profiling identified differential expression of genes encoding ECM components and enzymes required for ECM remodeling, such as matrix metalloproteinases (MMP) and ADAM endopeptidase enzymes, following manipulation of Pitx2c function. These potential Pitx2 targets have also been reported in other contexts, including during development of the zebrafish eye (**Hendee et al., 2018**). The hypothesis that Pitx2 regulates ECM dynamics has also been proposed in chick, where asymmetries in ECM and adhesion molecules acting downstream of Pitx2 regulate the asymmetric morphogenesis of the gut tube (**Kurpios et al., 2008**). Taken together, these data suggest that Pitx2c functions as a regulator of the expression of ECM components and remodelers to promote a particular environment conducive to cell migration during embryonic development.

Gastrulation movements occur within a dynamic environment. Hence, interaction between germ layers and with the surrounding matrix can influence individual cell behaviors. Our analysis of wild-type cells transplanted into MZ*pitx2c* mutant hosts shows that cell shape and protrusive behaviors are altered in the mutant environment. Coincident with these behaviors, we observed an aberrant fibronectin network and increased expression of *itgb1b*, which could disrupt focal adhesion formation and signaling, and cause cells to flatten over the underlying matrix. It is also interesting to consider how the ECM components may feed-forward to promote actin dynamics cell-autonomously. For example, genetic studies have shown that the hyaluronic acid synthase gene *has2* induces the

activation of Rac1 to promote lamellipodia formation and cell migration during zebrafish gastrulation (*Bakkers et al., 2004*). Protrusion stability and orientation can also be modulated by actin dynamics (*Davidson et al., 2006*; *Woo et al., 2012*) and PCP proteins (*Wallingford et al., 2000*; *Jessen et al., 2002*; *Ulrich et al., 2003*; *Kim et al., 2010*). Interestingly, crosstalk occurs between PCP signaling and the ECM, as PCP components can regulate the deposition and remodeling of the ECM (*Dzamba et al., 2009*; *Tao et al., 2009*; *Dohn et al., 2013*; *Huang and Kornberg, 2016*), while mechanical strain can affect planar polarization (*Chien et al., 2015*). It will be interesting to elucidate the hierarchy of these events in cell migration during gastrulation, as well as investigate whether these pathways are conserved in other developmental contexts where Pitx2c functions, including the establishment of left-right asymmetry.

# Materials and methods

## Key resources table

| Reagent type (species) or resource | Designation | Source or reference | Identifiers | Additional information |
|---|---|---|---|---|
| Genetic reagent (*Danio rerio*) | *pitx2c^ups6* | This paper | | |
| Genetic reagent (*D. rerio*) | *Tg(sox17:GFP)^s870* | doi.org/10.1016/j.devcel.2008.02.012 | ZFIN ID: ZDB-ALT-061228–2 | |
| Sequence-based reagent | sox32 morpholino | doi.org/10.1016/S0925-4773 (01)00453–1 | ZFIN ID: ZDB-MRPHLNO-050818–2 | |
| Antibody | anti-GFP (chicken) | Aves Lab | GFP_1020 | |
| Antibody | anti-Fibronectin (rabbit) | Sigma | F3648 | |
| Antibody | anti-phospho-Histone H3 (Ser10) (rabbit) | Millipore | 06–570 | |
| Antibody | anti-cleaved Caspase 3 (Asp175) (rabbit) | Cell Signaling | | |
| Antibody | Alexa 488-, 568-, or −647 secondaries | Molecular Probes | | |
| Commercial assay or kit | mMESSAGE mMACHINE SP6 Transcription Kit | ThermoFisher Scientific | AM1340 | |
| Commercial assay or kit | DIG RNA labeling kit | Roche | 11277073910 | |
| Commercial assay or kit | RNA Clean and Concentrator kit | Zymo Research | R1013 | |
| Commercial assay or kit | Maxima First Strand cDNA Synthesis Kit for RT-qPCR with DNAse | ThermoFisher Scientific | K1641 | |
| Commercial assay or kit | DyNAmo ColorFlash SYBR Green qPCR kit | ThermoFisher Scientific | F416 | |
| Software | ZEN Blue 2012 | Zeiss, Germany | | |
| Software | ZEN Black 2012 | Zeiss, Germany | | |
| Software | NIS-Elements | Nikon Instruments Inc. | | |
| Software | Imaris - Version 8.4.0 | Bitplane, UK | | |
| Software | FIJI/ImageJ | doi:10.1038/nmeth.2019 | | |
| Software | GraphPad Prism 7 | GraphPad Software, California, USA | | |

## Zebrafish husbandry

Zebrafish husbandry was performed under standard conditions in accordance with institutional (MPG) and national ethical and animal welfare guidelines. Procedures involving animals were

approved by the veterinary department of the Regional Board of Darmstadt and the French veterinary service and ethical committee (ID: A-31-555-01 and APAPHIS #3653–2016011512005922 v6).

## Generation of mutant alleles of *pitx2c* and genotyping

A pair of TALE nucleases was assembled as previously described (*Huang et al., 2011*). For each nuclease, a target specific TALE DNA binding domain consisting of 16 RVD repeats was constructed from plasmids containing single RVD repeats, which were kindly provided by Bo Zhang (Peking University, China). The target sequences for the TALE DNA-binding domains were selected in the first coding exon of the *pitx2c* specific isoform just upstream of the 3' exon-intron junction and flanking an ApaL1 site to facilitate screening for indels in the F0. The sequence of the targeted fragment is (TALE binding sites are indicated in bold and the ApaL1 site is underlined): T**CGAAACACAGGC TGGA**T<u>GTGCAC</u>ACGGTTTCA**GACACCTCCAGTCCAG**A

The assembled TALE repeats were subcloned in a pCS2 vector containing appropriate Δ152 Nter TALE, +63 Cter TALE, and FokI cDNA sequences with the appropriate half-TALE repeat (derived from the original pCS2 vector (*Huang et al., 2011*). TALEN expression vectors were linearized by NotI digestion and capped RNAs were synthesized using the mMESSAGE mMACHINE SP6 Kit (Life Technologies).

Screening for indels in the F0 and F1 was performed using the following primers and the resulting PCR fragment was digested with ApaL1:

pitx2c_F1 gtatttgtgacctgcgctga
pitx2c_R1 tgaagcccacattttcacaa

From the F2 onwards, genotyping of the resulting Δ4 allele was performed using high resolution melt analysis with the following primers:

HRMA_pitx2c_F1: cttcagcggtctgtggactc
HRMA_pitx2c_R1: acctacggactctggactgg

The F1 generation was outcrossed at least once before raising homozygous mutant adult fish. Homozygous mutant fish were raised from heterozygous incrosses. Experiments were performed using females from the F3 generation onwards; phenotypic analyses were performed on F4-F7 generation animals.

## In situ hybridization

In situ hybridization was performed as described in (*Thisse and Thisse, 2008*). Digoxygenin-labelled anti-sense probes were synthesized using T7 polymerase (Roche) and DIG RNA labelling kit (Roche). The sequences of primers used to amplify probe templates are listed in *Supplementary file 2*. Embryos were imaged on a Nikon SMZ25 stereomicroscope.

## Mosaic labelling of notochord cells

mRNAs encoding membrane *tdTomato* and *H2B-GFP* were synthesized using the mMESSAGE mMACHINE SP6 kit (Ambion). Primer sequences for cloning are listed in *Supplementary file 3*. 50–100 pg of each mRNA were injected into a single blastomere at the 32–64 cell stage to label only a subset of cells. Embryos were incubated at 28.5°C until they reached the two somite stage (ss). Images were processed, and the cell aspect ratio of labelled cells was measured in FIJI/ImageJ.

## Photoconversion of lateral mesendodermal cells

Wild-type and MZ*pitx2c* mutant embryos were injected at the 1 cell stage with 100 pg of photoconvertible *kikGR* mRNA and raised until shield stage. Primer sequences for cloning are listed in *Supplementary file 3*. Embryos were positioned with the lateral side facing the objective and the shield at 90°. A 100 $\mu m^2$ region was selected for photoconversion with a 405 nm wavelength laser. Images of successfully photoconverted embryos were acquired at 70 and 90% epiboly. To calculate anterior extension, the length of the photoconverted cell population was measured in at least eight embryos. To ascertain the extent of dorsal convergence, the angle between the photoconverted cell population to the dorsal midline was calculated. Photoconversion and image acquisition was performed on a Zeiss LSM700 confocal microscope, and measurements were performed using FIJI/ImageJ.

## Endodermal migration imaging and quantification

The *pitx2c Δ4* allele was crossed into the *Tg(sox17:GFP)$^{s870}$* line (*Chung and Stainier, 2008*) to image endodermal cell migration. Images were acquired every 3 min for 60-90 min using a Zeiss Spinning Disk confocal microscope, and the embryos were oriented with the lateral side facing the objective and the shield at 90°. Images were processed in ImarisTrack (Bitplane), and tracks were manually edited. Track speed, track displacement and track straightness parameters were quantified from at least 20 tracks per 3–4 embryos.

## Cell division imaging

To assess angles of cell divisions, embryos were injected with 100 pg of membrane *tdTomato* and nuclear *H2B-GFP* mRNA at the 1 cell stage. Primer sequences for cloning are listed in *Supplementary file 3*. Embryos were incubated at 28.5°C until they reached shield stage. Images were acquired every 3 min for 60 min using a Zeiss Spinning Disk confocal microscope, and the embryos were oriented with the lateral side facing the objective and the shield at 90°. Angles were measured using FIJI software and binned into groups using GraphPad. Chi-square tests were performed using GraphPad. Individual angles were converted to radians and input into MatLab to generate rose diagrams.

## Immunohistochemistry

Immunohistochemistry was performed according to standard protocols. Fibronectin staining was performed according to a protocol provided by Scott Holley. Stained embryos were mounted in 1% low-melt agarose and imaged on an LSM800 confocal microscope or Spinning Disk confocal microscope (Zeiss). Primary antibodies used were chicken anti-GFP (1:200, Aves Labs), rabbit anti-phospho-Histone H3 (1:2000, Millipore), rabbit anti-cleaved Caspase-3 (1:200, Cell Signaling), mouse anti-Fibronectin (1:100, Sigma). Secondary antibodies (Life Technologies) were used at 1:300. Positive cells were counted using FIJI/Image J.

## Cell transplantation

Wild-type donor embryos were injected at the 1 cell stage with one or a combination of the following: FITC dextran, rhodamine dextran, 100 pg membrane *tdTomato* mRNA, 100 pg *LIFEACT-GFP* mRNA, and 100 pg *ndr2* mRNA. Embryos were dechorionated using pronase for 5 min in 1/3 Ringer solution supplemented with penicillin (50 U/ml) and streptomycin (50 μg/ml). Cells were taken from donor embryos and transplanted at the blastoderm margin of wild-type or MZ*pitx2c* mutant hosts at dome and imaged after 3 hr (~60% epiboly) and/or after 6 hr (~1–2 ss). Embryos were not sorted subsequently based on phenotype classification, therefore MZ*pitx2c* mutant hosts and donors exhibited a combination of phenotypic severity. For live imaging experiments, successfully transplanted embryos were imaged from shield until tailbud on a Zeiss Spinning Disk confocal microscope. Images were acquired every 90 s. Images were rendered, and cell surface areas and volumes were calculated using Imaris software (Bitplane).

## mRNA overexpression

Full length *pitx2c* coding sequences were amplified from wild-type and *pitx2c$^{-/-}$* cDNA using primers listed in *Supplementary file 3*. Amplicons were cloned into pCS2 linearized plasmid using Cold Fusion (System Biosciences) and sequenced to confirm correct inserts. Plasmids were linearized and transcribed using the mMESSAGE mMACHINE SP6 kit (Ambion). RNA was purified using the RNA Clean-up and Concentrator Kit (Zymo Research). For all gain-of-function experiments, 25 pg of *pitx2c* mRNA was injected at the 1 cell stage for global overexpression, unless otherwise specified.

## Microarray expression profiling

For gain-of-function analysis, total RNA was isolated from a pool of 20 dome stage embryos following global *pitx2c* overexpression (25 pg) as well as a pool of 20 dome stage uninjected control embryos. For loss-of-function analysis, total RNA was isolated from a pool of 20 shield stage MZ*pitx2c* mutant embryos as well as a pool of 20 stage-matched *pitx2c$^{+/-}$* control embryos. Sample quality was tested using a Bioanalyzer. Microarray analysis was performed by Oak Labs (Germany). Dataset analysis was performed using DAVID Functional Annotation Tool (*Huang et al., 2009a*;

*2009b*) using gene lists generated by filtering genes for a fold-change >2 (LOF dataset) or >2.5 (GOF dataset) and removing genes detected below a cutoff of 10. Gene IDs that were unmapped by DAVID (e.g. genscan IDs) were excluded from analysis. Datasets have been deposited on GEO with the accession number GSE114671.

## RT-qPCR

Total RNA was isolated from pools of 20 embryos from the appropriate genotypes and at the relevant stages using Trizol (Life Technologies). RNA was purified using the RNA Clean-up and Concentrator Kit (Zymo Research). cDNA synthesis was performed using the Maxima First Strand cDNA Synthesis kit for RT-qPCR with DNAse treatment (Thermo Fisher) according to manufacturer's instructions. Diluted cDNA was used in qPCR reactions with DyNAmo colorFlash SYBR green qPCR kit using the CFX Connect Real-Time system (Bio-Rad). mRNA expression levels were normalized to *rpl13*. Average Ct values for all conditions are listed in *Supplementary file 1*. Sequences of qPCR primers are listed in *Supplementary file 4*.

## Statistics

All statistics were performed using GraphPad software. No prior analysis was done to calculate sample size.

## Acknowledgements

We would like to thank Sébastien Gauvrit, Rashmi Priya, Chi-Chung Wu, Felix Gunawan, and Anna Sokol for discussion and comments on the manuscript, the reviewers for constructive feedback, Marianne Ploch and Rebecca I-Ching Lee for technical assistance, Radhan Ramadass for microscopy and MatLab advice, Rita Retzloff and Martin Laszczyk for zebrafish care, and Scott Holley for sharing his fibronectin staining protocol. This work was supported by grants from the Fondation pour la Recherche Médicale (DEQ20131029166, PB), Deutsches Zentrum Für Herz Kreislauf Forschung (DZHK) (MMC), and the Max Planck Society (DYRS).

## Additional information

### Competing interests

Didier YR Stainier: Senior editor, *eLife*. The other authors declare that no competing interests exist.

### Funding

| Funder | Grant reference number | Author |
| --- | --- | --- |
| Max-Planck-Gesellschaft | | Didier YR Stainier |
| Deutsches Zentrum für Herz-Kreislaufforschung | | Michelle M Collins |
| Fondation pour la Recherche Médicale | DEQ20131029166 | Patrick Blader |

The funders had no role in study design, data collection and interpretation, or the decision to submit the work for publication.

### Author contributions

Michelle M Collins, Conceptualization, Data curation, Formal analysis, Funding acquisition, Investigation, Methodology, Writing—original draft, Writing—review and editing; Hans-Martin Maischein, Investigation, Performed transplantation experiments; Pascale Dufourcq, Resources, Generated the pitx2c line; Marine Charpentier, Resources, Generated TALEN plasmid; Patrick Blader, Resources, Writing—review and editing; Didier YR Stainier, Conceptualization, Formal analysis, Supervision, Funding acquisition, Writing—original draft, Writing—review and editing

Author ORCIDs
Michelle M Collins ⓘ http://orcid.org/0000-0002-8439-5278
Patrick Blader ⓘ http://orcid.org/0000-0003-3299-6108
Didier YR Stainier ⓘ http://orcid.org/0000-0002-0382-0026

## Ethics

Animal experimentation: This study was performed in strict accordance with institutional (MPG) and national ethical and animal welfare guidelines. Procedures involving animals were approved by the veterinary department of the Regional Board of Darmstadt (Permit Number: B2/Anz.1001) and the French veterinary service and ethical committee (ID: A-31-555-01 and APAPHIS #3653-2016011512005922v6).

## Decision letter and Author response

Decision letter https://doi.org/10.7554/eLife.34880.040
Author response https://doi.org/10.7554/eLife.34880.041

# Additional files

## Supplementary files

• Supplementary file 1. Ct values of genes by RT-qPCR.
DOI: https://doi.org/10.7554/eLife.34880.032

• Supplementary file 2. Primers for probe amplification.
DOI: https://doi.org/10.7554/eLife.34880.033

• Supplementary file 3. Primers for cloning.
DOI: https://doi.org/10.7554/eLife.34880.034

• Supplementary file 4. qPCR primer sequences.
DOI: https://doi.org/10.7554/eLife.34880.035

• Transparent reporting form
DOI: https://doi.org/10.7554/eLife.34880.036

## Data availability

Microarray analyses have been deposited in GEO under accession code GSE114671. All data generated or analysed during this study are included in the manuscript and supporting files. Source data files have been provided for all figures showing quantification.

The following dataset was generated:

| Author(s) | Year | Dataset title | Dataset URL | Database, license, and accessibility information |
|---|---|---|---|---|
| Collins, MM, Stainier, DYR | 2018 | Pitx2c regulates axis extension via mesendodermal cell migration | https://www.ncbi.nlm.nih.gov/geo/query/acc.cgi?acc=GSE114671 | Publicly available at the NCBI Gene Expression Omnibus (accession no: GSE114671) |

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
