## [Decision Letter]

Thank you for submitting your article "Pitx2c orchestrates embryonic axis extension via mesendodermal cell migration and oriented cell division" for consideration by *eLife*. Your article has been reviewed by three peer reviewers, including Lilianna Solnica-Krezel as the Reviewing Editor and Reviewer #1and the evaluation has been overseen by Anna Akhmanova as the Senior Editor.

The reviewers have discussed the reviews with one another and the Reviewing Editor has drafted this decision to help you prepare a revised submission. We hope you will be able to submit the revised version within two months.

Summary:

Collins and colleagues report essential role of Pitx2c transcription factor in convergence and extension movements during zebrafish gastrulation. Whereas the essential and conserved role of Pitx2 in left-right patterning and organogenesis is known, here the authors discover that Pit2c is also maternally expressed in addition to its zygotic expression largely in the nascent mesoderm and later prechordal plate mesoderm. Surprisingly, maternal zygotic *pitx2c* mutants exhibit morphologic defects during gastrulation that the authors attribute to abnormal morphogenesis, and in particular defective convergence and extension movements. At the cellular level, the authors correlate the abnormal CE movements with less persistent mesendodermal cell migration, abnormal cell shape and defective polarity of cellular divisions. They further attribute these defects to transcriptional changes in components of the chemokine-ECM-integrin signaling pathway that has been previously implicated in endodermal cell migration during gastrulation. They propose a non cell-autonomous role of Pitx2 in this process.

This work is significant as it advances our understanding of gastrulation. Whereas several signaling pathways, such as Nodal, and Cxcl12b-Cxcr4 have been implicated in aspects of gastrulation, transcriptional regulation of these movements is still very poorly understood.

This manuscript significantly expands our understanding of gastrulation by defining a role of an isoform of *pitx2* in coordinating morphogenesis during gastrulation. The significant conceptual advances proposed in this manuscript include: (1) Pitx2c is required during gastrulation to control cell behavior, (2) Pitx2c acts non-autonomously during gastrulation and (3) this early role of Pitx2c is mediated by downstream target genes, including the Cxcl12b chemokine, the integrin β subunit 1b, and components of the PCP pathway (*wdpcp* and *frzb*). This role is likely to be conserved since, as the authors state, early *pitx2* expression is also found in the mouse and chick. The work is generally of high quality and well presented.

However, some conclusions are not fully supported by the current data, and interpretation of some results is problematic. These issues need to be fully addressed before the manuscript becomes suitable for publication.

Essential revisions:

1) A more comprehensive characterization of the gastrulation phenotype is needed. It is concluded that MZ*pitx2c* mutants exhibit morphogenetic defects without defects in patterning or cell fate specification. Yet, in Figure 1 bilateral somitic myoD1 staining is fused in the midline, suggesting axial mesoderm deficiency. Axial/chordamesoderm markers are also presented in Figure 2 without obvious gaps. The conclusion of normal patterning in MZ*pitx2c* requires further evidence. Can gene expression profiling shed some light on this question?

2) Are cell death and/or proliferation contributing factors to the observed morphogenetic defects? The mutant embryo with severe CE defects presented in Figure 1e seems to exhibit cell death. And in some images, there appear to be fewer cells in the mutants. If so, in addition to defective cell movements during gastrulation, Pitx2 may regulate cell proliferation and cell death, these cell behaviors need to be examined here to rule out such possibility.

3) Along the same lines, Pitx2 has been involved in differential cell sorting and regulates N-cadherin expression (Plageman, 2011; Kurpios et al., 2008). It is possible that Pitx2 may be regulating differential cell adhesion in zebrafish. This possibility should be at least considered in the context of the gene expression profiling data.

4) Convergence movements' analysis needs to be properly performed. To measure the dorsal convergence of mesendoderm populations, the authors photoconverted cells in embryos injected with a photoconvertible *kikGR* RNA and measured the anteroposterior extension and width of the photoconverted population. This strategy is appropriate for analyzing extension in zebrafish, but not to measure convergence. In zebrafish dorsal convergence is mostly driven by dorsal migration of mesendodermal cells, thus a translocation of converted population towards the dorsal midline, rather than mediolateral narrowing of the labeled population itself should be assessed.

5) Cell-autonomy experiments as currently presented are flawed and need to be carried out with an appropriate readout at appropriate time. The authors transplanted into wild-type hosts at 30% epiboly two populations of cells: wild-type cells and MZ*pitx2c* mutant cells labeled with different fluorescent markers. They state, "After 3 hr post-transplant, we did not observe any differences in the two labelled populations." There are two problems with this conclusion. First, there is no quantification of these results. Second and a fundamental one, is that this assessment was done about 7 hours post fertilization (30% epiboly + 3 hours), whereas the earlier analyses of cell migration in MZ*pitx2c* revealed no significant defects of mesendodermal cell migration during early gastrulation. Neither the authors have observed any defects with the random walk of endoderm. Rather it was the second phase of endodermal migration convergence, during late gastrulation that appeared affected. Therefore, the conclusion that "Pitx2 likely does not function cell-autonomously within the context of cell migration" is not supported by these experiments. To address the issue of cell-autonomy effectively, an assessment of these experiments should be carried out at late gastrulation, e.g. yolk plug closure. Moreover, other cellular defects, like cell shape and protrusive activity, could be investigated as a readout in these cell autonomy studies.

6) The proposed mechanism downstream of Pitx2c involving Cxcl12b-Cxcr4 signaling also requires additional experimental assessment. In images of *sox17* expression marking endodermal and dorsal forerunner cells shown in Figure 2G, the authors note abnormal appearance of dorsal forerunner cells that appear scattered and fail to coalesce. It is surprising that the authors do not comment on the pattern of *sox17* expression in endodermal cells, as abnormal anterior migration of endodermal cells and in particular their abnormal location closer to the animal pole, is a hallmark of defective Cxcl12b-Cxcr4 signaling reported by the Schilling laboratory (Science 2008). As the authors propose misregulation of Cxl12b as the main mechanism downstream of Pitx2, *sox17* expression should be more closely examined. Likewise, what do endoderm derivatives look like in MZ*pitx2c* mutants at 1dpf? Can any aspects of the phenotype be rescued by restoring normal Cxcl12b expression? This is an important point, given the above questions about sox17 expression pattern in MZ*pitx2* mutant. Currently, the proposed relationship between Pitx2c, Cxcl12b and abnormal cell movements is merely a correlation.

7) The role of impaired cell division polarity is also not clear, especially the role of defective PCP signaling. The partial rescue of the orientation of cell divisions in MZ*pitx2c* mutants by downstream target genes is not very convincing, despite the Chi-squared analysis. This is particularly true of the *vangl2* rescue, shown in Figure 8F.

---

## [Author Response]

Essential revisions:1) A more comprehensive characterization of the gastrulation phenotype is needed. It is concluded that MZpitx2c mutants exhibit morphogenetic defects without defects in patterning or cell fate specification. Yet, in Figure 1 bilateral somitic myoD1 staining is fused in the midline, suggesting axial mesoderm deficiency. Axial/chordamesoderm markers are also presented in Figure 2 without obvious gaps. The conclusion of normal patterning in MZpitx2c requires further evidence. Can gene expression profiling shed some light on this question?

To further investigate patterning of MZ*pitx2c* mutants, we have analyzed the expression levels of dorsal organizer genes (n=4) and mesendodermal markers (n=20) in our two datasets. In agreement with previous reports (Essner et al., 2000; Faucourt et al., 2001), overexpression of Pitx2c can affect the expression levels of many genes (e.g., *dkk1a* and *chd*). However, in MZ*pitx2c* mutants, the expression level of the vast majority of genes was unchanged at shield stage, including those encoding key transcription factors that regulate germ layer fate (e.g., *sox32, gata5*) and signaling molecules (e.g., *ndr1, ndr2*). These data further support our interpretation that early embryonic patterning is not strongly affected in the maternal zygotic mutant embryos. We have now included a new figure with these data (Figure 7—figure supplement 2A) and have described these observations within the Results section and Discussion section.

2) Are cell death and/or proliferation contributing factors to the observed morphogenetic defects? The mutant embryo with severe CE defects presented in Figure 1e seems to exhibit cell death. And in some images, there appear to be fewer cells in the mutants. If so, in addition to defective cell movements during gastrulation, Pitx2 may regulate cell proliferation and cell death, these cell behaviors need to be examined here to rule out such possibility.

To address this possibility, we performed phospho-Histone H3 immunostaining at late gastrulation and observed a significant reduction in the number of proliferating cells in MZ*pitx2c* mutant embryos, which may indeed contribute to the phenotype. To address cell death, we performed cleaved Caspase-3 immunostaining at late gastrulation; however, no significant changes were observed between wild-type and MZ*pitx2c* embryos. We now have included these data in Figure 2—figure supplement 1 with the corresponding description in the text.

3) Along the same lines, Pitx2 has been involved in differential cell sorting and regulates N-cadherin expression (Plageman, 2011; Kurpios et al., 2008). It is possible that Pitx2 may be regulating differential cell adhesion in zebrafish. This possibility should be at least considered in the context of the gene expression profiling data.

To determine whether Pitx2c can regulate differential cell adhesion, we looked for any transcriptional changes in key adhesion molecules in our microarray data. Indeed, our DAVID pathway analysis identified that a subset of genes upregulated in MZ*pitx2c* mutant embryos were categorized as ‘cell adhesion’ genes (now in revised Figure 7—figure supplement 1). We expanded our analysis for additional cell adhesion genes and have listed them in a revised figure (Figure 7—figure supplement 2C). These data have also been described in the Discussion section of the text.

4) Convergence movements' analysis needs to be properly performed. To measure the dorsal convergence of mesendoderm populations the authors photoconverted cells in embryos injected with a photoconvertible kikGR RNA and measured the anteroposterior extension and width of the photoconverted population. Whereas this strategy is appropriate for analyzing extension in zebrafish, but not to measure convergence. In zebrafish dorsal convergence is mostly driven by dorsal migration of mesendodermal cells, thus a translocation of converted population towards the dorsal midline, rather than mediolateral narrowing of the labeled population itself should be assessed.

We thank the reviewers for this critical feedback and have now re-analyzed our data and performed additional experiments. We have changed Figure 3 accordingly, and now present two graphs with the quantification of anterior extension and the angle of dorsal convergence of photoconverted lateral mesendodermal cells. Furthermore, we have updated the Materials and methods section to reflect these new analyses.

5) Cell-autonomy experiments as currently presented are flawed and need to be carried out with an appropriate readout at appropriate time. The authors transplanted into wild-type hosts at 30% epiboly two populations of cells: wild-type cells and MZpitx2c mutant cells labeled with different fluorescent markers. They state, "After 3 hr post-transplant, we did not observe any differences in the two labelled populations." There are two problems with this conclusion. First, there is no quantification of these results. Second and a fundamental one, is that this assessment was done about 7 hours post fertilization (30% epiboly + 3 hours), whereas the earlier analyses of cell migration in MZpitx2c revealed no significant defects of mesendodermal cell migration during early gastrulation. Neither the authors have observed any defects with the random walk of endoderm. Rather it was the second phase of endodermal migration convergence, during late gastrulation that appeared affected. Therefore, the conclusion that "Pitx2 likely does not function cell-autonomously within the context of cell migration" is not supported by these experiments. To address the issue of cell-autonomy effectively, an assessment of these experiments should be carried out at late gastrulation, e.g. yolk plug closure. Moreover, other cellular defects, like cell shape and protrusiveactivity, could be investigated as a readout in these cell autonomy studies.

We thank the reviewers for these important comments and agree that the most relevant time point had not been analysed. We have now performed live imaging of these cell transplants and analysed additional parameters of the transplantation experiments at late gastrulation (70-90% epiboly): transplanted cell protrusions, transplanted cell surface area and volume measurements, and quantification of the migration distance of transplanted cells along the anterior-posterior axis. We have also analysed these parameters at the late gastrulation to tailbud stage. These data have now been included in a revised Figure 6 and new Figure 6—figure supplement 1. These new experiments support the conclusion that the role of Pitx2c during mesendodermal migration is cell non-autonomous.

6) The proposed mechanism downstream of Pitx2c involving Cxcl12b-Cxcr4 signaling also requires additional experimental assessment. In images of sox17 expression marking endodermal and dorsal forerunner cells shown in Figure 2G, the authors note abnormal appearance of dorsal forerunner cells that appear scattered and fail to coalesce. It is surprising that the authors do not comment on the pattern of sox17 expression in endodermal cells, as abnormal anterior migration of endodermal cells and in particular their abnormal location closer to the animal pole, is a hallmark of defective Cxcl12b-Cxcr4 signaling reported by the Schilling laboratory (Science 2008). As the authors propose misregulation of Cxl12b as the main mechanism downstream of Pitx2, sox17 expression should be more closely examined. Likewise, what do endoderm derivatives look like in MZpitx2c mutants at 1dpf? Can any aspects of the phenotype be rescued by restoring normal Cxcl12b expression? This is an important point, given the above questions about sox17 expression pattern in MZpitx2c mutant. Currently, the proposed relationship between Pitx2c, Cxlcl12b and abnormal cell movements is merely a correlation.

We agree with the reviewers that the proposed relationship between *pitx2c* and *cxcl12b* is correlative at this time. We have tried to perform rescue experiments by injecting *cxcl12b* mRNA into MZ*pitx2c* mutant embryos; however, these injections led to severe defects that precluded the analysis of endodermal rescue. Given that *cxcl12* does not appear to be expressed ubiquitously, it is challenging to restore the wild-type expression pattern and level. We have however carried out additional analysis of endodermal derivatives in MZ*pitx2c* mutants at 24 hpf, which revealed that the anterior endoderm remains much wider in mutants compared to wild types. Images and quantification of the width of the anterior endoderm are now shown in Figure 4.

7) The role of impaired cell division polarity is also not clear, especially the role of defective PCP signaling. The partial rescue of the orientation of cell divisions in MZpitx2c mutants by downstream target genes is not very convincing, despite the Chi-squared analysis. This is particularly true of the vangl2 rescue, shown in Figure 8F.

We have decided to remove these data from the manuscript, as we agree that they are not very convincing. We have kept the data showing that the orientation of cell division was disrupted in the MZ*pitx2c* mutants and have discussed it in more detail in the context of current literature within the Discussion section to address this concern.